# STAR-KV: Low-Rank KV Cache Compression via Soft Thresholding for Adaptive Rank Control

Priyansh Bhatnagar [* 1]   Ashkan Moradifirouzabadi [* 1]   Se-Hyun Yang [2]   SeungJae Lee [2]   Jungwook Choi [3]   Mingu Kang [1]

## Abstract

Low-rank projection has emerged as a promising approach for compressing the KV cache by exploiting hidden-dimension redundancy. However, prior methods rely on fixed or heuristic rank selection and struggle to achieve aggressive compression with minimal accuracy degradation. We propose STAR-KV, an adaptive low-rank KV cache compression framework with fine-grained rank control. STAR-KV encompasses 1) a differentiable thresholding mechanism that enables optimal rank selection at both attention-head and block levels, 2) a hybrid decomposition strategy that applies different low-rank factorizations according to the sensitivity of key and value projections, and 3) a low-rank–aware mixed precision quantization that leverages data statistics for near lossless low-bit quantization. Evaluated across multiple LLMs and benchmarks, STAR-KV achieves up to 75% KV cache compression and up to 20× overall KV cache reduction when combined with quantization. Enabled by custom Triton-based GPU kernels, STAR-KV delivers up to 6.9× speedup for the attention module and 3.1× end-to-end generation throughput. Our code is publicly available at: https://github.com/PriyanshBhatnagar/STAR-KV.

## 1. Introduction

Recent advances in large language models (LLMs) are driven by rapid scaling in both model size and supported context length. Conventional models support context windows on the order of $10^5$ tokens, such as up to 128K (Meta;

*Equal contribution  [1]University of California San Diego, San Diego, CA, USA [2]Dnotitia Inc., Seoul, Republic of Korea [3]Hanyang University, Seoul, Republic of Korea. Correspondence to: Mingu Kang <mingu@ucsd.edu>.

*Proceedings of the $43^{rd}$ International Conference on Machine Learning*, Seoul, South Korea. PMLR 306, 2026. Copyright 2026 by the author(s).

Grattafiori et al., 2024), while state-of-the-art work demonstrates architectures capable of handling context lengths of one million tokens or more (Liu et al., 2024b; Ding et al., 2024). However, supporting longer contexts introduces severe system-level bottlenecks during inference (Kwon et al., 2023; Hooper et al., 2024). A primary source of these bottlenecks is the key–value (KV) cache, whose storage and access costs increase as generation proceeds. The KV cache stores key and value tensors associated with previously processed tokens to avoid recomputation, resulting in a memory footprint that grows linearly with context length and dominates GPU memory consumption and bandwidth, often exceeding the storage required for model weights (Dao, 2023; Gholami et al., 2024). For example, when fully utilizing the 128K token context window in LLaMA-3.1-8B (Grattafiori et al., 2024) with a batch size of four, the KV cache accounts for 81% of the model's total 85GB FP16 memory footprint, becoming a central obstacle to long-context inference.

Recently, KV cache compression using *low-rank decomposition* has emerged as a promising direction that targets redundancy in the hidden (channel) dimension of key and value tensors. Low-rank decomposition faces two fundamental challenges. First, aggressive compression introduces *information loss* whose impact varies substantially across layers and attention heads, making it difficult to maintain model quality at high compression rates. As a result, prior work (Chang et al., 2025; Yan et al., 2025; Saxena et al., 2024) typically limits compression to moderate levels (e.g., 30–50%), since pushing beyond this regime leads to noticeable quality degradation. Second, although compression reduces memory footprint, reconstructing compressed key and value tensors at inference introduces *additional computation*, which significantly increases inference latency (Chang et al., 2025; Yan et al., 2025; Saxena et al., 2024).

To address these challenges, we introduce STAR-KV (Soft Thresholding for Adaptive Rank KV Cache Compression), a framework for KV cache compression via fine-grained rank control. To enhance the accuracy–compression trade-off, STAR-KV proposes an *adaptive rank learning* framework that reduces the latent dimensionality of low-rank KV representations at fine granularities. Specifically, we parameter-

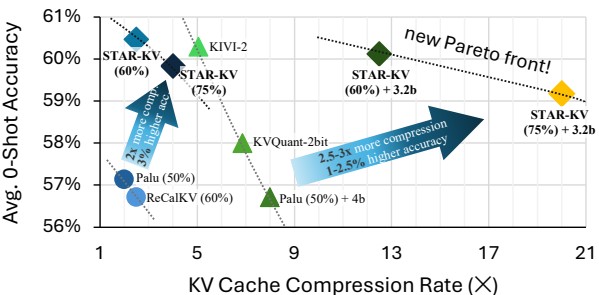

*Figure 1.* The average zero-shot accuracy of LongChat-v1.5-7B on five tasks (ARC-e, ARC-c, OBQA, PIQA, Hella) across different KV cache compression rates applied by SoTA methods.

ize the rank selection of each low-rank KV representation using a differentiable threshold applied to the singular values. By doing so, STAR-KV yields rank profiles that are learned, in contrast to prior approaches that set ranks via heuristics or sensitivity estimations (Saxena et al., 2024; Chang et al., 2025; Zhang et al., 2024).

Furthermore, low-rank decomposition can be applied at different granularities: (i) *head-wise decomposition* (HD), which factorizes each head's projection independently, and (ii) *joint decomposition* (JD), which factorizes a single matrix formed by concatenating heads. These two decomposition granularities exhibit trade-offs in terms of accuracy and reconstruction overhead. We characterize this trade-off along with a sensitivity analysis and adopt a *hybrid decomposition* scheme in STAR-KV: (i) HD for the key projection to minimize reconstruction overhead, and (ii) JD for the value projection to better preserve fidelity.

Finally, STAR-KV adopts a *mixed-precision low-rank–aware quantization* that synergistically leverages the inherently ordered singular values of the key/value projection matrices to demarcate important channels from the rest. This enables efficient identification and handling of outlier channels in contrast to prior KV quantization methods that rely on explicit outlier modeling, non-uniform quantization, or additional computations to address outliers at high cost (Hooper et al., 2024; Liu et al., 2024c; Su et al., 2025; Ashkboos et al., 2024; Tseng et al., 2024).

Evaluated on a wide range of LLMs and tasks, STAR-KV yields improved *accuracy–compression* Pareto trade-offs relative to prior KV cache compression approaches, as shown in Fig. 1, achieving up to **4.0× low-rank KV cache compression** with a negligible accuracy degradation compared to the uncompressed model, and 2.70% better average zero-shot accuracy and 2.0× higher compression rate compared to a SoTA method, Palu (Chang et al., 2025). When combined with 3.2-bit average mixed-precision quantization, STAR-KV attains up to **20.0× overall KV cache compression** while achieving a higher zero-shot accuracy by 1.18% and 2.47% and compression by 2.5× and 2.9×

over KVQuant (Hooper et al., 2024) and Palu with 4-bit quantization, respectively. With custom Triton-based GPU kernels, STAR-KV (75%) achieves up to **6.9×** and **4.0× speedup in the attention operator** compared to PyTorch SDPA implementation (Shah et al., 2024), with and without 4-bit quantization, respectively. STAR-KV also yields **3.1× higher end-to-end generation throughput** (tokens/second) in long context (128K) or high batch size scenarios compared to PyTorch native implementation. In summary, our contributions are:

- A *learnable and adaptive* low-rank KV cache compression method that automatically determines optimal ranks at decoder block and attention head granularity.
- A *hybrid decomposition* scheme that exploits the differing sensitivity of key and value tensors under compression with minimal reconstruction overhead.
- A *low-rank–aware mixed-precision quantization* strategy that leverages the data statistics induced by low-rank compression.
- Customized *Triton-based GPU kernels* that translate the proposed techniques into real speedups.

**Conflict of Interest Disclosure.** Se-Hyun Yang and SeungJae Lee are employed by Dnotitia Inc., which develops technologies related to this work. Mingu Kang has a financial interest in Dnotitia Inc.

## 2. Related Work

**KV Cache Optimization.** Recent attention variants, such as grouped-query attention (GQA) (Ainslie et al., 2023) and multi-head latent attention (MLA) (Liu et al., 2024a), reduce KV cache memory by sharing or projecting key/value heads. However, these approaches require architectural changes and additional (re-)training, limiting their applicability to existing models. In contrast, lightweight post-training alternatives include KV cache quantization, which reduces key/value precision (Hooper et al., 2024; Liu et al., 2024c), and token eviction, which retains only a subset of tokens under a cache budget (Zhang et al., 2023; Xiao et al., 2024).

**Low-Rank KV Cache.** Low-rank KV cache methods exploit redundancy in the hidden (channel) dimension of key/value tensors by projecting them into a lower-dimensional latent space (Chang et al., 2025; Saxena et al., 2024; Lin et al., 2024; Yan et al., 2025). Unlike quantization or token eviction, these methods cache compact latent representations. A common approach is *low-rank decomposition*, which factorizes key/value projection weights offline during model preparation (Chang et al., 2025). The resulting low-rank factors are used directly at inference time, incurring no additional runtime overhead.

# 3. Background

## 3.1. Multi-Head Attention (MHA)

Attention mechanism (Vaswani et al., 2017) selectively focuses on relevant sections of the input sequence using three separate spaces: query, key, and value. Given an input $x$, these spaces are computed as, $Q = x \cdot W_Q^T$, $K = x \cdot W_K^T$, $V = x \cdot W_V^T$, where $W_Q$, $W_K$, and $W_V$ denote the projection matrices. For head $i$, the attention weights are computed as $P^i = \text{Softmax}\left(\frac{Q^i \cdot K^{iT}}{\sqrt{d_h}}\right)$, where $d_h$ is the per-head dimensionality. The output of multi-head attention is obtained by computing the weighted sum for each head, concatenating the results, and projecting back to the original hidden dimension: $\text{MHA}(x) = \left(P^i \cdot V^i\right)_{\text{concat}_i} \cdot W_{out}^T$, where $W_{out}$ is the output projection matrix.

## 3.2. Low-rank KV Cache

Singular Value Decomposition (SVD) factorizes a matrix $W \in \mathbb{R}^{m \times n}$ as $W = U\Sigma V^\top$, where $U \in \mathbb{R}^{m \times m}$ and $V \in \mathbb{R}^{n \times n}$ are orthogonal and $\Sigma \in \mathbb{R}^{m \times n}$ is diagonal with singular values in descending order; a rank-$r$ approximation keeps the top-$r$ components, yielding $W \approx U_r \Sigma_r V_r^\top$ with $U_r \in \mathbb{R}^{m \times r}$, $\Sigma_r \in \mathbb{R}^{r \times r}$, and $V_r \in \mathbb{R}^{n \times r}$. The key and value projection matrices $(W_K, W_V \in \mathbb{R}^{d_{out} \times d_{in}})$ are factorized using SVD for low-rank KV cache compression, and only the top-$r$ singular values and corresponding singular vectors are retained for a rank-$r$ approximation. During inference, the truncated decompositions of the key and value projection matrices are expressed in factorized form, where the right singular vectors and singular values are merged into a single matrix as $W_K \approx U_K (\Sigma V)_K^T$ and $W_V \approx U_V (\Sigma V)_V^T$, where $U_K, U_V \in \mathbb{R}^{d_{out} \times r}$ and $(\Sigma V)_K^T, (\Sigma V)_V^T \in \mathbb{R}^{r \times d_{in}}$ denote the low-rank factors of the key and value projections, respectively, yielding the corresponding low-rank key representation as $K = (x \cdot (\Sigma V)_K) \cdot U_K^T = K' \cdot U_K^T$, and value representation as $V = (x \cdot (\Sigma V)_V) \cdot U_V^T = V' \cdot U_V^T$. In this formulation, the intermediate key $(K')$ and value $(V')$ representations lie in a reduced $r$-dimensional subspace and are stored in the KV cache instead of the full-dimensional tensors, resulting in a reduced memory footprint.

## 3.3. Weight Absorption

A key technique for reducing the computational complexity in low-rank KV caching is *weight absorption*, introduced in DeepSeek-V2 (Liu et al., 2024a) in the context of MLA. The core idea is to absorb the reconstruction matrices (e.g., $U_K$ and $U_V$) into the query and output projections (e.g., $W_Q$ and $W_{out}$) by reordering associative matrix multiplications, so that reconstruction cost is reduced at decoding time. However, recent LLMs apply Rotary Positional Embedding (RoPE (Su et al., 2024)) to the query and key states

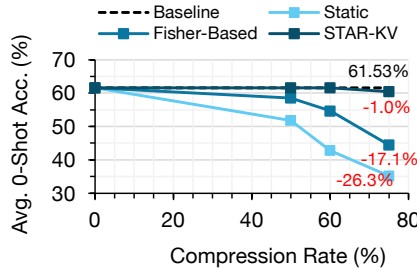

*Figure 2.* Comparison of rank selection strategies at varying KV cache compression rates. Average zero-shot accuracy is reported for LongChat-v1.5-7B evaluated on common sense tasks.

prior to computing attention scores. The position-dependent RoPE matrix is inserted between $W_Q$ and $U_K$, preventing offline fusion (Li et al., 2025; Ji et al., 2025). In contrast, value processing does not involve positional encoding and therefore allows operation reordering. The attention output computation for head $i$, $(p^i \cdot V^i) \cdot W_{out}^{iT}$, is reordered as: $\left(p^i \cdot V' \cdot U_V^{iT}\right) \cdot W_{out}^{iT} = (p^i \cdot V') \cdot \left(U_V^{iT} W_{out}^{iT}\right)$, where $p^i \in \mathbb{R}^l$ are the attention probabilities, $V' \in \mathbb{R}^{l \times r_v}$ are the compressed value states, $U_V^i \in \mathbb{R}^{d_h \times r_v}$ is the value reconstruction matrix, and $W_{out}^i \in \mathbb{R}^{d \times d_h}$ is the head-wise output projection. Before weight absorption, the total FLOPs of the value processing across $h$ heads is $l \cdot r_v \cdot h^2 \cdot d_h + l \cdot h \cdot d_h$. By defining the precomputed $W_{out}'^i \triangleq W_{out}^i U_V^i$, the reordering of operations reduces the FLOPs count to $l \cdot r_v \cdot h \cdot d_h + h \cdot r_v \cdot d_h$. With a sufficiently large sequence length $l$, the computational saving approaches a factor of $d_h$, which is typically 128 in current widely used LLMs.

# 4. The STAR-KV Framework

## 4.1. Rank-Adaptive KV Cache Compression

### 4.1.1. Rank Selection Challenge for KV Cache

Low-rank KV cache compression critically depends on *rank allocation* across decoder blocks, as latent dimensionality directly controls the trade-off between approximation error and memory reduction. Aggressive rank reduction degrades accuracy, whereas conservative choices limit KV cache savings. Selecting ranks that balance this trade-off under a global compression budget is therefore a central challenge. Rank allocation largely follows two paradigms. (1) *Static rank assignment* uses a uniform rank (or fixed compression ratio) across blocks and often across $W_K$ and $W_V$, ignoring heterogeneous block-wise and operator sensitivity. (2) *Heuristic-based allocation* assigns ranks using offline importance proxies (e.g., Fisher-information- scores (Ly et al., 2017)) (Chang et al., 2025). While these heuristics improve over uniform ranks at moderate compression, they exhibit non-negligible accuracy loss at high compression budgets. Fig. 2 shows degradation increases with compression rate,

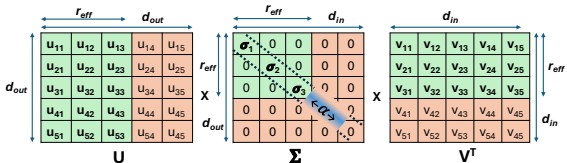

*Figure 3*. A projection matrix is decomposed as $W = U\Sigma V^T$. Applying a threshold $\alpha$ on the diagonal of $\Sigma$ suppresses singular values with $\sigma_i < \alpha$ to zero and retains only the corresponding columns of $U$ and rows of $V_T$, yielding an effective rank $r_{\text{eff}}$.

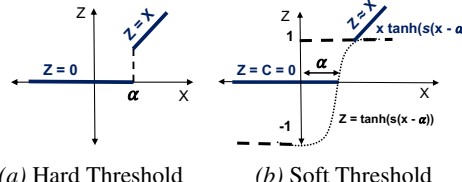

*(a)* Hard Threshold      *(b)* Soft Threshold

*Figure 4*. Thresholding operators: (a) conventional non-differentiable hard threshold, and (b) soft threshold using shifted tanh with sharpness control factor $s$.

reaching 26.33% for the static and 17.14% for the heuristic-based approach at 75% compression. This motivates an adaptive rank-selection strategy that supports aggressive KV cache compression while minimizing performance loss.

### 4.1.2. ADAPTIVE RANK SELECTION VIA SOFT THRESHOLDING

To address the rank selection challenge, STAR-KV introduces an adaptive rank selection method which applies a threshold to singular values of KV projection matrices.

**Thresholding Singular Values.** We parameterize rank selection via a threshold $\alpha$ applied to the singular spectrum, which directly controls the *effective rank* ($r_{\text{eff}}$) of each decomposed projection. Given the SVD, $W = U\Sigma V^\top$ with $\Sigma = \text{diag}(\sigma_1, \ldots, \sigma_p)$ and $\sigma_1 \geq \cdots \geq \sigma_p \geq 0$, we define the thresholded spectrum

$$\widetilde{\Sigma}_\alpha \triangleq \text{diag}(\tilde{\sigma}_1, \ldots, \tilde{\sigma}_p), \qquad \tilde{\sigma}_i \triangleq \sigma_i \mathbf{1}\{\sigma_i \geq \alpha\}. \quad (1)$$

As shown in Fig. 3, increasing $\alpha$ moves the cutoff upward along the diagonal of $W_\Sigma$, decreasing the number of retained singular values and reducing the effective rank $r_{\text{eff}}$, while decreasing $\alpha$ retains more components increasing $r_{\text{eff}}$.

**Soft Threshold Mechanism.** To capture the varying rank sensitivity across decoder blocks, the threshold must be learned independently for the key and value projections in each block. In singular value thresholding, optimizing the threshold parameter $\alpha$ implicitly controls the effective rank $r_{\text{eff}}$; however, hard thresholding is non-differentiable (Fig. 4a) and therefore unsuitable for gradient-based optimization. (Li et al., 2022) proposes a differentiable thresholding mechanism in the context of token pruning by replacing the hard threshold with a surrogate function. Inspired by it, we apply a differentiable thresholding operator (Fig. 4b) on the singular values. Let $x$ denote a singular value (a diagonal entry of $W_\Sigma$). We define the soft-threshold operator

$$\mathcal{T}h_s(x; \alpha) = \begin{cases} x \tanh\big(s(x - \alpha)\big), & x \geq \alpha, \\ c \tanh\big(s(x - \alpha)\big), & x < \alpha, \end{cases} \quad (2)$$

where $\alpha$ is a learnable threshold, $s > 0$ controls transition sharpness, and $c$ is a constant (we set $c = 0$ so sub-threshold values are suppressed). This operator is

a smooth surrogate for spectral truncation: for $x \gg \alpha$, $\tanh(s(x - \alpha)) \to 1$ and the singular value is approximately preserved; for $x < \alpha$, the output is driven toward zero. Larger $s$ yields a closer approximation to a hard cutoff, while smaller $s$ produces a smoother transition. Applying $\mathcal{T}h_s(\cdot; \alpha)$ to the singular spectra of the key/value projections yields $W_{K_{\Sigma_{r_{\text{eff}}}}} = \mathcal{T}h_s(W_{K_\Sigma}; \alpha_K)$ and $W_{V_{\Sigma_{r_{\text{eff}}}}} = \mathcal{T}h_s(W_{V_\Sigma}; \alpha_V)$, where $\alpha_K$ and $\alpha_V$ are learned independently (per block and attention head). As discussed in Section 5.1, this training process is done on a small subset of data, taking only <6 GPU hours. Once training is completed, we employ hard thresholding offline prior to inference and replace the decomposed projection matrices with their truncated form during inference.

### 4.2. Adaptive Compression Objective

To maximize low-rank KV cache compression while preserving model performance, we optimize a joint objective that balances compression and accuracy through an adaptive compression loss and a knowledge distillation loss.

**Compression Loss.** Each decoder block contains learnable soft-threshold parameters that control compression for the key and value projections. Let $\alpha_i$ denote the threshold parameter associated with decoder block $i$. We introduce an adaptive compression loss

$$\mathcal{L}_{\text{acmp}} = \sum_i e^{-\alpha_i}. \quad (3)$$

This formulation naturally yields phase-adaptive compression behavior. When $\alpha_i$ is small, $e^{-\alpha_i}$ is large, producing a strong optimization signal that quickly increases $\alpha_i$ and thus boosts compression in the early stage of training. As $\alpha_i$ grows, the exponential term decays rapidly, leading to diminishing compression pressure. This slows the growth of $\alpha_i$ and allows the model to focus on recovering and stabilizing task performance at the achieved compression level.

**Knowledge Distillation Loss.** We compute the distillation loss using the Kullback–Leibler divergence:

$$\mathcal{L}_{\text{KD}} = \text{KL}\big(\mathbf{p}_t \parallel \mathbf{p}_s\big), \quad (4)$$

where $\mathbf{p}_t$ and $\mathbf{p}_s$ denote the output token distributions of the teacher and student models, respectively. This encourages

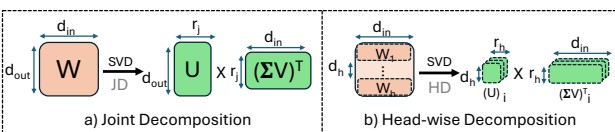

*Figure 5.* Comparison of JD and HD.

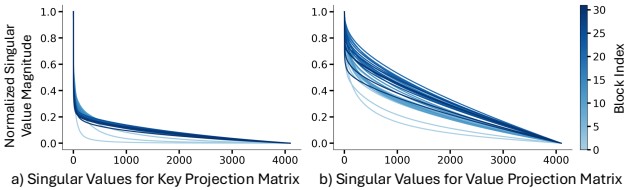

a) Singular Values for Key Projection Matrix    b) Singular Values for Value Projection Matrix

*Figure 6.* The singular-value spectra of the key and value projection matrices in LLaMA-2-7B, normalized by their spectral norms ($\|\cdot\|_2$), across 32 decoder blocks.

the compressed student model to match the teacher's predictive distribution, preserving model behavior under aggressive compression. The final training objective is therefore:

$$\mathcal{L}_{\text{tot}} = \mathcal{L}_{\text{KD}} + \gamma \cdot \mathcal{L}_{\text{acmp}}, \tag{5}$$

where $\gamma$ scales the contributions of the compression loss, thereby modulating their relative influence on the objective.

### 4.3. Decomposition Scheme

Fig. 5 shows low-rank decomposition can be applied at different granularities: (i) *joint decomposition* (JD), which factorizes a single matrix formed by concatenating heads, and (ii) *head-wise decomposition* (HD), which factorizes each head's projection independently. We analyze the trade-off in terms of accuracy and reconstruction overhead to propose a decomposition scheme for STAR-KV in this section.

#### 4.3.1. RECONSTRUCTION OVERHEAD ANALYSIS

In JD, all heads are concatenated and reconstructed jointly using a shared cache $A \in \mathbb{R}^{l \times r_j}$ and reconstruction matrix $B \in \mathbb{R}^{r_j \times d}$, where $r_j = c\,d$, $d = h\,d_h$, $c$ is the compression rate, and $d_h$ is the per-head dimension. The reconstruction cost is $l \cdot r_j \cdot d = l \cdot c \cdot h^2 \cdot d_h^2$. In contrast, HD reconstructs each head independently with per-head cache $A_h \in \mathbb{R}^{l \times r_h}$ and reconstruction matrix $B_h \in \mathbb{R}^{r_h \times d_h}$, where $r_h = c\,d_h$. Summing across $h$ heads gives $h \cdot l \cdot r_h \cdot d_h = l \cdot c \cdot h \cdot d_h^2$, which is a factor of $h$ lower than JD for the same compression rate.

**Observation 1.** The total reconstruction FLOPs in JD is $h \times$ larger than HD at the same compression rate $c$.

#### 4.3.2. RECONSTRUCTION ERROR ANALYSIS

In this section, we analyze the sensitivity of key and value projections by examining the singular value distributions of their corresponding projection matrices ($W_K$, $W_V$). We also analyze how different low-rank decomposition schemes affect the approximation error.

**Lemma 4.1.** *Let $W \in \mathbb{R}^{m \times n}$ have singular values $\sigma_1 \geq \cdots \geq \sigma_p$ with $p = \min(m, n)$, and let $W_r$ be its rank-$r$ truncated SVD. By Eckart-Young-Mirsky theorem, the relative error of the optimal rank-$r$ approximation in the spectral norm satisfies*

$$\varepsilon_2(W; r) \triangleq \min_{\text{rank}(X) \leq r} \frac{\|W - X\|_2}{\|W\|_2} = \frac{\sigma_{r+1}(W)}{\sigma_1(W)}. \tag{6}$$

*where $\|\cdot\|_2$ denotes the spectral norm ($\|W\|_2 = \sigma_1$).*

**Implication.** In Fig. 6, we plot the normalized spectra $\sigma_i/\sigma_1$ for $W_K$ and $W_V$. At the truncation index $r+1$, we observe that the value projection typically has a larger normalized singular value than the key projection, i.e.,

$$\frac{\sigma_{r+1}(W_K)}{\sigma_1(W_K)} \leq \frac{\sigma_{r+1}(W_V)}{\sigma_1(W_V)}. \tag{7}$$

**Observation 2.** (6) and (7) imply $\varepsilon_2(W_K; r) \leq \varepsilon_2(W_V; r)$, which indicates that under the same rank constraint, the optimal rank-$r$ approximation of values incurs higher (or equal) relative error than that of keys.

**Lemma 4.2.** *Let $W \in \mathbb{R}^{m \times d}$ be partitioned into $h$ heads along columns as $W = \begin{bmatrix} W^{(1)} & W^{(2)} & \cdots & W^{(h)} \end{bmatrix}$ with $W^{(i)} \in \mathbb{R}^{m \times d_h}$ for $i \in \{1, \ldots, h\}$ and $d = hd_h$. The best joint rank-$R$ approximation is*

$$\widehat{W}_{\text{joint}} \in \arg \min_{\text{rank}(X) \leq R} \|W - X\|_F, \tag{8}$$

*while the best head-wise approximation with rank at most $r$ per head is*

$$\widehat{W}_{\text{hw}} \in \arg \min_{\{X^{(i)} : \text{rank}(X^{(i)}) \leq r\}} \left\| W - \begin{bmatrix} X^{(1)} & \cdots & X^{(h)} \end{bmatrix} \right\|_F. \tag{9}$$

*Under the same compression budget $R = hr$, the joint reconstruction error is no larger than the head-wise error:*

$$\|W - \widehat{W}_{\text{joint}}\|_F \leq \|W - \widehat{W}_{\text{hw}}\|_F. \tag{10}$$

*Proof* for Lemma 4.2 is in Appendix A.2.

**Observation 3.** Under the same compression budget, the Frobenius-norm approximation error achieved by JD is no larger than that of HD.

#### 4.3.3. HYBRID KV DECOMPOSITION

The above observations indicate that (i) JD incurs $h \times$ higher reconstruction overhead than HD, (ii) value projections have higher (or equal) low-rank approximation error than key projections, and (iii) JD provides less (or equal) approximation error than HD. Fig. 7 further corroborates

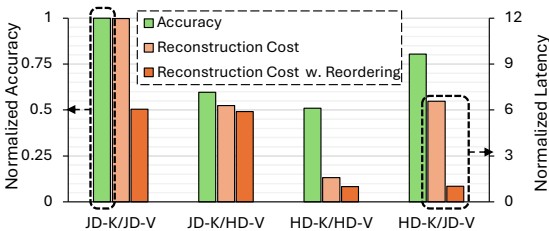

*Figure 7.* Normalized accuracy and reconstruction costs of JD and HD with different combinations for keys and values. The average zero-shot accuracy is measured across six tasks. The latency is the profiling result of the matrix multiplication operations involved in reconstruction on an NVIDIA RTX 4090 GPU.

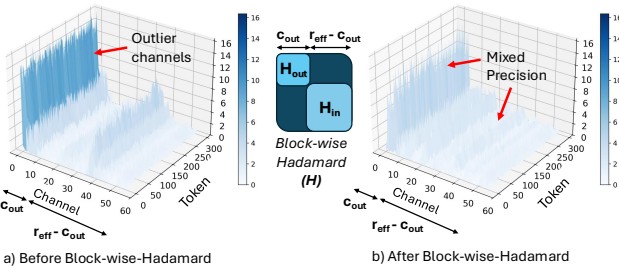

a) Before Block-wise-Hadamard                    b) After Block-wise-Hadamard

*Figure 8.* Visualization of low-rank key states for the first head of the third layer in LongChat before and after applying block-wise Hadamard, which redistributes channel magnitudes and enables mixed-precision quantization with minimal degradation.

this trend: applying JD to both $W_K$ and $W_V$ yields the best accuracy but incurs the highest reconstruction latency, whereas using HD for keys and JD for values achieves the best accuracy-overhead balance. Combining these insights, STAR-KV adopts a hybrid decomposition strategy: since values are more sensitive and JD attains lower approximation error, we apply JD to decompose value projections; conversely, we apply HD to the less sensitive key projections to reduce reconstruction overhead. Importantly, operation reordering is applicable to the value reconstruction (Section 3.3) to compensate for the overhead. As shown in Fig. 7, the normalized latency is substantially reduced for the HD-$K$/JD-$V$ configuration after reordering, making our hybrid decomposition optimal for maintaining accuracy-overhead balance. Consistent with our hybrid decomposition, we apply separate adaptive compression losses to keys and values to give finer-grained control by modifying (3):

$$\mathcal{L}_{\text{acmp}} = \sum_i \sum_h e^{-\alpha_{i,h}^K} + \sum_i e^{-\alpha_i^V}, \qquad (11)$$

where $\alpha_{i,h}^K$ denotes the threshold for the key projection of head $h$ in block $i$, and $\alpha_i^V$ denotes the threshold for the value projection shared across all the heads in block $i$.

## 4.4. Integration with Quantization

To further enhance compression beyond low-rank decomposition, we apply quantization to the resulting low-rank latent representations of keys and values. After decomposition, the latent key and value representations are scaled by the retained singular values. Because these singular values are ordered in decreasing magnitude, the resulting latent channels exhibit a highly skewed distribution: a small number of initial channels carry significantly larger magnitudes, whereas the remaining channels have much smaller magnitudes. This structure, illustrated in Fig. 8(a), introduces prominent outlier channels that make subsequent quantization particularly challenging. Prior work (Chang et al., 2025) mitigates this issue by applying a Hadamard transformation, which redistributes the outlier energy across channels. However, applying a Hadamard transform globally with uniform precision over all latent channels incurs substantial accuracy degradation (as shown in Appendix A.1.4), which becomes even more pronounced under aggressive compression. To address this limitation, we propose a low-rank–aware block-wise Hadamard quantization integrated with mixed precision. Let $Z \in \mathbb{R}^{l \times r}$ be the latent representation after low-rank compression. We partition the latent dimension into an outlier block $Z_{\text{out}}$ and an inlier block $Z_{\text{in}}$ based on channel magnitude, with $r_{\text{outlier}} + r_{\text{inlier}} = r$. As shown in Fig. 8, independent Hadamard transforms are applied to each block, followed by mixed-precision per-token quantization ($\mathcal{Q}$), given by:

$$\hat{Z} = \big[\mathcal{Q}_{b_{\text{out}}}(Z_{\text{out}}H_{\text{out}}) \quad \mathcal{Q}_{b_{\text{in}}}(Z_{\text{in}}H_{\text{in}})\big], \qquad (12)$$

where $H_{\text{out}}$ and $H_{\text{in}}$ are Hadamard matrices, $b_{\text{out}}$ and $b_{\text{in}}$ are the bit precision for the outlier and inlier blocks, respectively, and $b_{\text{out}} > b_{\text{in}}$ and $r_{\text{outlier}} \gg r_{\text{inlier}}$. Fig. 8(b) illustrates the two smoothed regions after applying block-wise Hadamard quantization. We reserve the top 20% of $r$ latent channels in $b_{out}$ precision and quantize the remaining 80% with $b_{in}$. Since $r$ is adaptive in our low-rank decomposition method, the corresponding block-wise Hadamard matrices also become adaptive to each decoder block (for keys and values) and to each head (for keys). Moreover, our block-wise Hadamard quantization incurs no overhead as the Hadamard factors can be fused into the decomposed up/down projections as $H^\top U^\top$ and $(V\Sigma)H$.

## 5. Experiments

### 5.1. Experiment Settings

**Models and Tasks.** We evaluate our method on four LLM model families: LongChat-7B-v1.5-32K (Li et al., 2023), LLaMA-2-7B (Touvron et al., 2023), LLaMA-3-8B-Instruct (Grattafiori et al., 2024), and LLaMA-3.1-8B-Instruct (Grattafiori et al., 2024). Model performance is evaluated using perplexity on WikiText-2 (Merity et al.,

| Model | Comp (%) | Perplexity ↓ | | | Zero-Shot Accuracy (%) ↑ | | | | | | |
|---|---|---|---|---|---|---|---|---|---|---|---|
| | | Wiki2 | C4 | lm-avg. | OBQA | PIQA | ARC-e | ARC-c | Hella | Wino | Avg. |
| **LongChat-7B-v1.5** | 0 | 6.86 | 9.93 | 8.40 | 41.00 | 76.28 | 71.84 | 41.38 | 71.20 | 67.48 | 61.53 |
| Palu | 50 | 7.37 | 11.67 | 9.52 | 38.20 | 73.78 | 67.63 | 37.63 | 68.43 | 65.27 | 58.49 |
| ReCalKV | 70 | 9.01 | 13.63 | 11.32 | 35.20 | 68.55 | 58.84 | 33.53 | 63.18 | 59.12 | 53.07 |
| STAR-KV | 60 | 6.83 | 9.98 | 8.41 | 41.80 | 75.90 | 72.69 | 41.47 | 70.49 | 66.85 | 61.53 |
| STAR-KV | 75 | 7.34 | 10.44 | 8.89 | 42.60 | 74.86 | 71.63 | 41.55 | 68.52 | 63.77 | 60.49 |
| **LLaMA-2-7B** | 0 | 5.12 | 7.04 | 6.08 | 44.20 | 78.07 | 76.30 | 46.42 | 76.00 | 69.30 | 65.05 |
| Palu | 50 | 5.63 | 8.38 | 7.01 | 43.60 | 76.33 | 73.02 | 42.57 | 73.39 | 66.67 | 62.60 |
| ReCalKV | 70 | 6.75 | 13.63 | 11.32 | 39.80 | 74.48 | 70.37 | 39.42 | 69.59 | 65.75 | 59.90 |
| STAR-KV | 60 | 5.49 | 7.45 | 6.47 | 43.40 | 79.22 | 75.55 | 44.71 | 74.63 | 68.51 | 64.34 |
| STAR-KV | 75 | 5.86 | 8.02 | 6.94 | 43.00 | 78.18 | 73.78 | 41.64 | 73.14 | 66.61 | 62.73 |
| **LLaMA-3-8B-Inst** | 0 | 7.74 | 12.61 | 10.18 | 43.00 | 78.51 | 81.61 | 56.57 | 75.95 | 71.59 | 67.87 |
| Palu | 50 | 9.43 | 17.37 | 13.40 | 42.40 | 76.12 | 76.14 | 49.06 | 70.33 | 71.82 | 64.31 |
| STAR-KV | 60 | 8.52 | 13.51 | 11.01 | 42.20 | 78.13 | 79.29 | 49.83 | 73.37 | 69.69 | 65.42 |
| STAR-KV | 75 | 10.20 | 15.46 | 12.83 | 42.20 | 78.02 | 77.57 | 49.32 | 71.59 | 66.22 | 64.15 |

*Table 1.* Zero-shot evaluation on commonsense reasoning benchmarks using LM-Eval-Harness. We report accuracy (%) under different KV cache compression rates. Perplexity on WikiText-2 and C4 is reported separately.

| Model | Comp (%) | Qasper | QMSum | TriviaQA | MultiQA | TREC | MultiNews | VCSum | Avg. |
|---|---|---|---|---|---|---|---|---|---|
| **LongChat-7B-v1.5** | 0 | 27.7 | 22.7 | 82.3 | 43.0 | 68.5 | 26.1 | 15.4 | 40.81 |
| Palu | 30 | 23.81 | 22.64 | 80.61 | 44.15 | 64.5 | 25.4 | 14.08 | 39.31 |
| Palu | 50 | 21.1 | 22.4 | 75.81 | 40.78 | 62.5 | 22.6 | 12.58 | 36.8 |
| STAR-KV | 60 | 22.58 | 22.4 | 81.3 | 41.04 | 65.0 | 25.7 | 15.6 | 39.09 |
| **LLaMA-3.1-8B-Inst** | 0 | 25.19 | 23.2 | 92.00 | 39.90 | 72.5 | 26.9 | 15.91 | 42.23 |
| Palu | 30 | 14.47 | 23.3 | 86.71 | 26.57 | 73.0 | 26.19 | 8.33 | 36.93 |
| Palu | 50 | 15.38 | 22.10 | 73.36 | 27.58 | 63.5 | 21.66 | 1.95 | 32.21 |
| STAR-KV | 60 | 23.2 | 22.4 | 89.06 | 36.34 | 67.0 | 25.89 | 13.2 | 39.58 |

*Table 2.* LongBench evaluation for STAR-KV and Palu under different KV cache compression rates.

2016) and C4 (Dodge et al., 2021b), and zero-shot accuracy measured with LM-Evaluation-Harness (Gao et al., 2024) on six tasks. For long-context evaluation, we additionally benchmark our method on LongBench (Bai et al., 2024) and RULER (Hsieh et al., 2024), which together cover diverse document-level and structured long-context tasks. Additional dataset and setting details are in Appendix A.4.

**Training and Compression Settings.** To learn optimal thresholds for adaptive low-rank decomposition, we fine-tune models on a calibration set of merely 3000 samples from FineWeb-Edu (Lozhkov et al., 2024) dataset using our custom loss described in Section 4.2, and evaluate our method under two KV cache compression budgets - 60% and 75%. Additional details are provided in Appendix A.3. Due to the small calibration set, the fine-tuning process incurs a modest training cost, taking just <6 GPU hours in total while using NVIDIA RTX Pro 6000.

**GPU Kernel Implementation.** To efficiently execute attention in STAR-KV, we implement custom kernels in Triton (Tillet et al., 2019). These kernels fuse bandwidth-bound element-wise operations (e.g., RoPE and dequantization) with the matrix multiplications used for reconstruction,

thereby reducing intermediate materialization and memory traffic. The details of the Triton kernels and additional implementation optimizations are provided in Appendix A.6.

### 5.2. Accuracy Analysis

We show that adaptive rank selection across blocks and heads, enabled by the soft thresholding, achieves significantly higher accuracy than uniform rank allocation, as demonstrated in Appendix A.1.1 and Fig. 2. We further study the accuracy impact across various benchmarks by comparing state-of-the-art adaptive rank control methods.

**Perplexity and Zero-Shot Evaluation.** Table 1 shows our method consistently preserves model quality under high KV cache compression rates. On LongChat-7B-v1.5, our method at 60% compression achieves a WikiText-2 perplexity of 6.83, matching the uncompressed baseline and outperforming Palu at lower compression rates, while also substantially improving over ReCalKV (Yan et al., 2025). Similar trends are observed on LLaMA-2-7B and LLaMA-3-8B-Instruct, where our method maintains lower perplexity at both 60% and 75% compression compared to Palu and ReCalKV. We further evaluate zero-shot accuracy using

| Model / Method | Comp (%) | MK1 | MK2 | MQ | MV | S1 | S2 | S3 | FWE | SQ | Avg. |
|---|---|---|---|---|---|---|---|---|---|---|---|
| **LongChat-7B-v1.5** | 0 | 99.80 | 99.60 | 98.40 | 96.55 | 100.0 | 100.0 | 100.0 | 48.20 | 56.72 | 88.80 |
| Palu | 50 | 99.80 | 98.80 | 75.30 | 97.40 | 100.0 | 100.0 | 98.60 | 42.40 | 53.25 | 85.06 |
| STAR-KV | 60 | 99.40 | 99.60 | 98.50 | 96.55 | 100.0 | 100.0 | 99.80 | 46.00 | 54.45 | 88.26 |
| STAR-KV, diff. seed | 60 | 99.20 | 99.80 | 98.35 | 97.60 | 100.0 | 100.0 | 100.0 | 49.93 | 53.48 | 88.71 |
| **LLaMA-3.1-8B-Inst** | 0 | 100.0 | 99.8 | 99.9 | 98.95 | 100.0 | 100.0 | 99.6 | 96.07 | 78.12 | 96.94 |
| Palu | 50 | 98.60 | 99.80 | 75.35 | 69.65 | 99.80 | 96.60 | 88.40 | 85.13 | 58.58 | 85.06 |
| STAR-KV | 60 | 98.4 | 86.0 | 85.1 | 84.0 | 100.0 | 100.0 | 92.0 | 87.73 | 68.05 | 89.03 |

*Table 3.* RULER evaluation at a sequence length of 4K. All values are reported as percentages. Tasks evaluated are MK1/MK2: multi-key tasks, MQ: multi-query, MV: multi-value, S1–S3: single-key tasks, FWE: few-shot word extraction, and SQ: SQuAD-style QA.

LM-Eval-Harness across six benchmarks. Our method consistently achieves higher accuracy and compression than Palu and ReCalKV. On LongChat-7B-v1.5, our method at 60% compression completely recovers the average baseline accuracy. For LLaMA-2-7B and LLaMA-3-8B-Instruct STAR-KV shows minimal degradation at 60% compression and remains competitive by outperforming both Palu and ReCalKV even at 75% compression.

**Long Context Evaluation.** To evaluate long-context performance, we compare STAR-KV against Palu on both Long-Bench and RULER, and report the results in Table 2 and Table 3, respectively. Maintaining long-context accuracy becomes challenging at higher compression rates, particularly for GQA-based models, a phenomenon also observed by Palu. As shown in Table 2, Palu exhibits noticeable degradation at 50% compression and therefore adopts a lower compression rate of 30% for a better accuracy–compression trade-off. In contrast, STAR-KV maintains stronger accuracy at higher compression. At 60% compression, STAR-KV matches Palu on LongChat-7B-v1.5 and outperforms it on LLaMA-3.1-8B-Instruct, substantially narrowing the gap to the uncompressed baseline. We further validate this trend on RULER at a 4K sequence length. As shown in Table 3, STAR-KV consistently achieves higher compression than Palu while preserving stronger long-context retrieval accuracy. On LongChat-7B-v1.5, STAR-KV at 60% compression reaches an average score of 88.26, close to the dense baseline of 88.80 and higher than Palu at 50% compression. The additional run with a different seed obtains a higher average score of 88.71, indicating stable performance. On LLaMA-3.1-8B-Instruct, STAR-KV improves the RULER average from 85.06 to 89.03 over Palu at a higher compression rate. These results show that STAR-KV provides a more favorable accuracy–compression trade-off for long-context workloads across both LongBench and RULER.

**Quantization.** Table 4 presents the quantization results of STAR-KV. With our adaptive block-wise Hadamard quantization, we preserve the first 20% channels in 4-bit and quantize the remaining channels in 3-bit, resulting in average 3.2-bit precision. When applied on top of 75% compressed LongChat model using our low-rank decomposition, our quantization achieves 20× overall KV cache compression

| Method | Bit | Comp. | Avg. (%) |
|---|---|---|---|
| LongChat-7B-v1.5 | 16 | 1× | 60.34 |
| KIVI-2-gs32-r32 | 3.16 | 5.1× | 60.3 |
| KVQuant-2bit-1% | 2.33 | 6.9× | 58.00 |
| Palu (50%) | 3 | 10.4× | 56.42 |
| Palu (50%) | 4 | 8.0× | 56.71 |
| STAR-KV (60%) | 3.2 | 12.5× | 60.12 |
| STAR-KV (75%) | 3.2 | 20.0× | 59.18 |

*Table 4.* Quantization results on LongChat-7B-v1.5, reporting average KV bit precision, KV cache compression rate, and zero-shot accuracy on commonsense benchmarks (ARC-e, ARC-c, OBQA, PIQA, Hella).

while maintaining 59.18% 0-shot average accuracy. This substantially outperforms prior KV quantization methods in both compression and accuracy, including KIVI (Liu et al., 2024c), KVQuant (Hooper et al., 2024), and Palu, demonstrating strong synergy between our low-rank decomposition and block-wise Hadamard quantization. We further analyze the impact of different outlier–inlier splits and find that selecting the first 20% of channels as outliers yields the best accuracy–compression trade-off. Results are shown in Appendix A.1.6.

### 5.3. System-level Performance Analysis

We evaluate the end-to-end attention speedups by STAR-KV on LLaMA-2-7B, executed on a single NVIDIA RTX 4090, targeting regular consumer scenarios. We use PyTorch's FP16 `scaled_dot_product_attention` (SDPA) as the baseline, which dispatches to the fastest available kernel, including FlashAttention-2 (Dao, 2023) when applicable.

**Speedup.** Fig. 9 reports the speedup of the attention operator. We test two STAR-KV variants at 75% compression, with and without 4-bit KV quantization, and compare against Palu at 50% compression (Palu does not provide an open-source quantized variant, so we report its FP16 results only). All experiments use a batch size 16. Since the PyTorch baseline runs out of memory at 32K and 64K, we could not measure its latency on our hardware. We therefore estimated the baseline by linear extrapolation from the

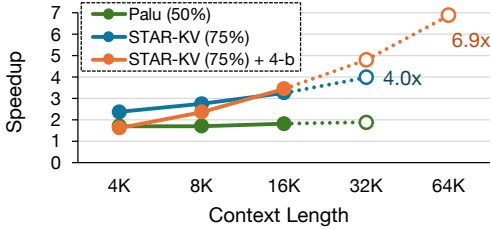

*Figure 9.* Normalized speedup of the attention module over the PyTorch SDPA FP16 implementation. The batch size is 16 in all setups. Solid lines represent exact measurements, while dashed lines indicate that the PyTorch baseline is out of memory. For these data points, the latencies are measured for the non-baseline variants, and the speedups are compared against the estimated baseline's latency.

| Contx. Len. | Pytorch SDPA | STAR-KV | Gain |
|---|---|---|---|
| 1K | 249.8 (16) | 751.0 (128) | 3.01× |
| 2K | 130.4 (8) | 400.7 (64) | 3.07× |
| 4K | 69.0 (4) | 212.2 (32) | 3.07× |
| 8K | 35.6 (2) | 110.6 (16) | 3.11× |
| 16K | 18.0 (1) | 56.6 (8) | 3.14× |

*Table 5.* End-to-end generation throughput (tokens/s). The gray text in brackets denotes batch size.

measured 8K and 16K latencies. We note, however, that the latency of STAR-KV, with and without quantization, is *measured on the GPU in all data points*, enabled by their reduced memory footprints. STAR-KV extends context length support to 32K while achieving a 4.0× speedup. Adding 4-bit KV quantization reduces speedup at short contexts due to quantization/dequantization overhead, but enables 64K context length with 6.9× speedup. A detailed breakdown of the attention latency is provided in Appendix A.5.5. To demonstrate the portability of our kernels, we extend the experiments to the RTX Pro 6000 GPU. When the context length is limited to fit within GPU memory, our method still achieves a 4.3× speedup. Details are provided in Appendix A.5.6.

**Generation Throughput.** We report the maximum achievable end-to-end generation throughput (tokens/s) of STAR-KV across context lengths in Table 5. For each context length, we increase batch size until the run becomes GPU-memory limited, and then measure the resulting generation throughput. As context length increases, the maximum feasible batch size decreases due to the larger KV cache footprint, which in turn reduces throughput. Overall, STAR-KV consistently improves end-to-end generation throughput across all tested settings, achieving a 3.1× average gain, with higher gains at longer context lengths. Notably, STAR-KV can fit up to 128K context length in a single RTX 4090.

## 6. Ablation Results

We present extended studies examining: 1) the sensitivity of the learned rank profiles to calibration datasets and nearby model variants, 2) the impact of the calibration dataset on the final compression–accuracy trade-off, 3) the sensitivity of STAR-KV to the compression-loss weight ($\gamma$) in the total training loss, and 4) the effect of combining STAR-KV with token eviction.

1) **Stability of learned rank profiles.** Our studies suggest that the learned rank profiles are highly consistent across different calibration datasets and transferable across nearby model variants, such as models from the same family with similar parameter counts. Results are shown in Appendix A.1.2.

2) **Sensitivity to the calibration dataset.** Our studies show that the learned thresholds are generally robust to the choice of calibration dataset, with at most $0.68\%$ difference in average zero-shot accuracy. Details are provided in Appendix A.1.3.

3) **Sensitivity to the compression-loss weight.** Our studies verify that the reported results are not caused by fragile hyperparameter settings. Using a different compression-loss weight results in only up to 0.25% average zero-shot accuracy variation. Details are shown in Appendix A.1.5.

4) **Combination with token eviction.** We study the impact of combining STAR-KV with H2O (Zhang et al., 2023), a representative KV cache compression method based on token pruning. Interestingly, on LongChat-v1.5-32K, combining 60% low-rank compression with 60% H20-style token pruning achieves 84% KV cache compression outperforming H2O with 60% pruning alone by 0.04% average zero-shot accuracy. Results are shown in Appendix A.5.4.

## 7. Conclusion

We present STAR-KV, an adaptive low-rank KV cache compression framework that enables fine-grained rank control across attention heads and decoder blocks. By introducing a soft-thresholding mechanism, STAR-KV learns effective rank allocation directly from data, avoiding static or heuristic rank selection. Guided by sensitivity analysis, STAR-KV adopts a hybrid decomposition strategy that balances reconstruction cost and accuracy, and further integrates a low-rank–aware mixed-precision quantization to enable near-lossless low-bit KV storage at high compression. Across multiple LLMs and benchmarks, STAR-KV achieves up to 75% KV cache reduction from low-rank compression alone and up to 20× overall KV cache reduction when combined with quantization, while maintaining model quality. Also, custom Triton kernels translate these algorithmic gains into practice, enabling 3.5× attention speedup at a context length of 16K and up to 6.9× (with an estimated baseline due to memory capacity limitation) at a context length of 64K. Finally, STAR-KV provides 3.1× end-to-end generation throughput improvement.

## Acknowledgments

This work was supported by the Korea Planning & Evaluation Institute of Industrial Technology (KEIT) grant funded by the Korea government (MOTIR) (No. RS-2024-00419977, Development of a Vector Database Accelerator for Large Language Models (LLM)).

This work was also supported by the National Research Foundation of Korea (NRF) grant funded by the Korea government (MSIT) (No. RS-2025-00561961).

## Impact Statement

This paper presents work whose goal is to advance the field of Machine Learning. There are many potential societal consequences of our work, none of which we feel must be specifically highlighted here.

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

# A. Appendix

## A.1. Ablation Studies

### A.1.1. STATIC VS. ADAPTIVE RANK SELECTION

We compare adaptive rank selection against static low-rank compression. Static rank selection assigns a *uniform* rank budget across all decoder blocks and attention heads, and applies the same rank to both key and value projections, ignoring the layer/head-specific spectral characteristics. As shown in Table 6, this uniform allocation leads to substantial accuracy degradation under the same compression target and training settings, highlighting the importance of STAR-KV's adaptive per-layer/head rank selection.

| Method | Comp (%) | OBQA | PIQA | ARC-e | ARC-c | Hella | Wino | Avg. |
|---|---|---|---|---|---|---|---|---|
| Static Low-rank Decomposition | 75 | 26.80 | 58.49 | 37.52 | 23.38 | 37.30 | 49.80 | 38.88 |
| STAR-KV | 75 | 42.60 | 74.86 | 71.63 | 41.55 | 68.52 | 63.77 | 60.49 |

*Table 6.* Comparison of static vs. adaptive rank selection under the same KV cache compression target.

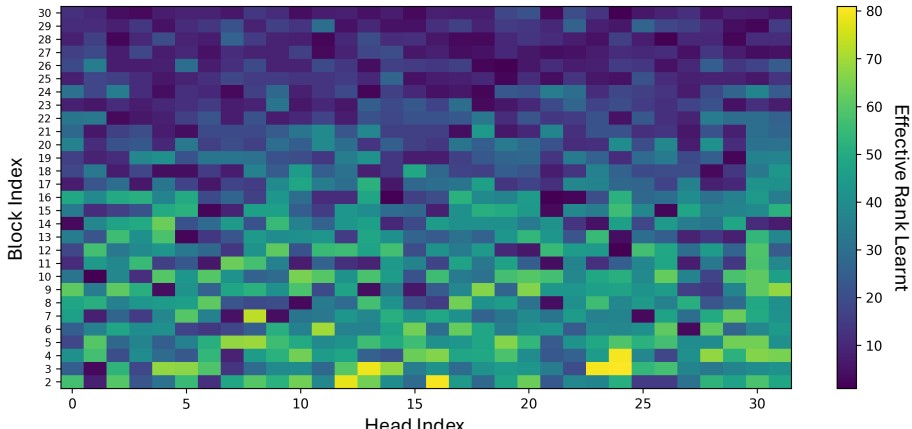

*Figure 10.* Ranks learned for Key projection matrices for each block and head using our adaptive rank selection strategy at 75% compression. The maximum rank of each head is 128.

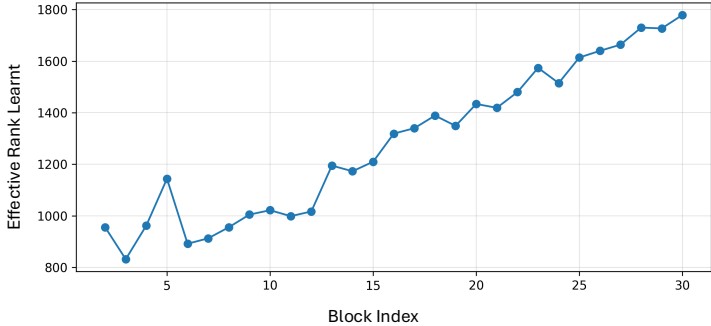

*Figure 11.* Ranks learned for Value projection matrices for each block using our adaptive rank selection strategy at 75% compression. The maximum rank of each layer is 4096.

Fig. 10 and Fig. 11 present the rank maps learned by the key and value projections, respectively, using our adaptive rank selection strategy based on the soft-thresholding mechanism.

A.1.2. STABILITY OF LEARNED RANK PROFILES

We further analyze whether the learned rank profiles are stable across calibration datasets and nearby model variants. To use a unified metric, we define a generic profile distance between two settings $A$ and $B$ as

$$\Delta_T(A, B) = \frac{1}{|\mathcal{I}_T|} \sum_{i \in \mathcal{I}_T} \left| p_i^T(A) - p_i^T(B) \right|, \qquad T \in \{K, V\}, \tag{13}$$

where $T$ denotes the tensor type, and $\mathcal{I}_T$ is the set of profile entries being compared. For keys, the profile is defined per layer and per head, i.e., $\mathcal{I}_K = \{(\ell, h) : \ell \in \mathcal{L}, h \in \mathcal{H}\}$; for values, the profile is defined per layer, i.e., $\mathcal{I}_V = \{\ell : \ell \in \mathcal{L}\}$. We instantiate $p_i^T(\cdot)$ differently depending on the comparison.

**Case 1: same model, different calibration datasets.** When comparing two calibration datasets $D$ and $D'$ for the same model, raw ranks are directly comparable. We therefore set

$$p_{(\ell,h)}^K(D) = r_{\ell,h}^K(D), \qquad p_\ell^V(D) = r_\ell^V(D), \tag{14}$$

where $r_{\ell,h}^K(D)$ is the learned key rank of head $h$ in layer $\ell$, and $r_\ell^V(D)$ is the learned value rank of layer $\ell$. Thus,

$$\Delta_K(D, D') = \frac{1}{|\mathcal{L}||\mathcal{H}|} \sum_{\ell \in \mathcal{L}} \sum_{h \in \mathcal{H}} \left| r_{\ell,h}^K(D) - r_{\ell,h}^K(D') \right|, \tag{15}$$

and

$$\Delta_V(D, D') = \frac{1}{|\mathcal{L}|} \sum_{\ell \in \mathcal{L}} \left| r_\ell^V(D) - r_\ell^V(D') \right|. \tag{16}$$

For readability, we report these rank differences together with their normalized forms, e.g., 1.6/128 ranks per head for keys and 20/4096 ranks per layer for values.

**Case 2: same calibration dataset, different model variants.** When comparing nearby model variants $M$ and $M'$ under the same calibration dataset, raw ranks may not be directly comparable due to different attention layouts, such as GQA and non-GQA. We therefore compare the learned compression-rate profiles instead of raw rank profiles. Specifically, for keys and values, we define the per-entry compression profile as the normalized learned rank:

$$c_{(\ell,h)}^K(M) = \frac{r_{\ell,h}^K(M)}{d_h}, \qquad c_\ell^V(M) = \frac{r_\ell^V(M)}{d_{\text{model}}}, \tag{17}$$

where $r_{\ell,h}^K(M)$ is the learned key rank of head $h$ in layer $\ell$, $r_\ell^V(M)$ is the learned value rank of layer $\ell$, $d_h$ is the key head dimension, and $d_{\text{model}}$ is the value hidden dimension. We then set

$$p_i^T(M) = c_i^T(M), \qquad T \in \{K, V\}. \tag{18}$$

The cross-model profile distance is defined as

$$\Delta_T(M, M') = \frac{1}{|\mathcal{I}_T|} \sum_{i \in \mathcal{I}_T} \left| c_i^T(M) - c_i^T(M') \right|, \qquad T \in \{K, V\}. \tag{19}$$

We report $\Delta_T(M, M')$ as a percentage by multiplying the normalized distance by 100.

| Comparison | Key-profile deviation | Value-profile deviation |
|---|---|---|
| C4 vs. FineWeb-Edu | 1.6/128 ranks per head (1.2%) | 20/4096 ranks per layer (0.5%) |
| RedPajama vs. FineWeb-Edu | 1.7/128 ranks per head (1.3%) | 7.7/4096 ranks per layer (0.2%) |
| LLaMA-2-7B vs. LLaMA-3.1-8B-Inst | 3.2% | 11.4% |

*Table 7.* Comparison of learned rank profiles across calibration datasets and nearby model variants. Dataset comparisons use LongChat-v1.5-32K.

Table 7 shows learned profiles are highly consistent across calibration datasets - FineWeb-Edu, C4 (Dodge et al., 2021a) and RedPajama (Weber et al., 2024). On LongChat-v1.5-32K, the key-rank difference is only 1.2–1.3% of the head dimension, while the value-rank difference remains below 0.5% of the model dimension. Across nearby model variants, the key compression profile remains especially stable, while the value profile shows larger but still moderate variation. This indicates that the learned rank allocation captures persistent layer-wise sensitivity patterns.

## A.1.3. SENSITIVITY TO THE CALIBRATION DATASET

We evaluate whether the choice of calibration dataset affects the final compression–accuracy trade-off. We calibrate STAR-KV using FineWeb-Edu, C4, and RedPajama, including an additional FineWeb-Edu run with a different seed. As shown in Table 8, the final zero-shot average varies only mildly across calibration datasets at the same 60% KV cache compression ratio, suggesting that the learned thresholds are generally robust to the calibration source.

| Model / Method | Calibration Data | Comp (%) | OBQA | PIQA | ARC-e | ARC-c | Hella | Wino | Avg. |
|---|---|---|---|---|---|---|---|---|---|
| **LongChat-v1.5-32K** | – | 0 | 41.00 | 76.28 | 71.84 | 41.38 | 71.20 | 67.48 | 61.53 |
| STAR-KV | FineWeb-Edu | 60 | 41.80 | 75.90 | 72.69 | 41.47 | 70.49 | 66.85 | 61.53 |
| | C4 | 60 | 42.20 | 76.77 | 71.76 | 40.78 | 70.53 | 66.30 | 61.39 |
| | RedPajama | 60 | 43.00 | 76.66 | 71.59 | 41.98 | 68.09 | 65.75 | 61.18 |
| | FineWeb-Edu, diff. seed | 60 | 42.60 | 76.06 | 71.93 | 41.81 | 68.54 | 65.98 | 61.15 |
| **LLaMA-3.1-8B-Inst** | – | 0 | 42.60 | 80.96 | 81.73 | 54.86 | 79.17 | 73.72 | 68.84 |
| STAR-KV | FineWeb-Edu | 60 | 41.80 | 79.27 | 81.02 | 51.45 | 75.52 | 69.53 | 66.43 |
| | C4 | 60 | 42.80 | 77.97 | 79.55 | 50.94 | 75.10 | 70.09 | 66.08 |
| | RedPajama | 60 | 42.40 | 78.07 | 79.46 | 50.43 | 74.37 | 69.77 | 65.75 |

*Table 8.* Sensitivity to calibration dataset at 60% KV cache compression.

## A.1.4. EFFECT OF HADAMARD ROTATION AND MIXED-PRECISION QUANTIZATION

| Method | Bit | OBQA | PIQA | ARC-e | ARC-c | Hella | Avg. |
|---|---|---|---|---|---|---|---|
| Global Hadamard (w/o mixed precision) | 3.0 | 40.80 | 74.32 | 69.32 | 39.42 | 67.27 | 58.23 |
| Mixed precision (w/o Hadamard) | 3.2 | 42.00 | 74.10 | 67.93 | 37.80 | 64.51 | 57.27 |
| STAR-KV | 3.2 | 41.80 | 74.86 | 70.83 | 40.70 | 67.72 | 59.18 |

*Table 9.* Ablation under 75% low-rank KV compression. We compare three strategies - 1) global Hadamard without mixed precision, 2) mixed-precision quantization without Hadamard transformation, and 3) the proposed block-wise Hadamard with mixed precision.

Table 9 isolates the roles of structured rotation and mixed-precision quantization. Applying a single global Hadamard with uniform 3-bit quantization improves robustness over naive quantization but still exhibits noticeable accuracy degradation, as it cannot selectively protect high-magnitude channels. Conversely, mixed-precision quantization alone, preserving the top 20% channels in 4-bit while quantizing the rest in 3-bit, resulting in a 3.2-bit average, also results in degraded performance, indicating that selective bit allocation without structured rotation is insufficient. In contrast, block-wise Hadamard enables effective mixed-precision quantization by redistributing outlier energy within each block. This combination achieves the best overall accuracy, demonstrating that neither Hadamard rotation nor mixed precision alone is sufficient; their synergy becomes vital for robust low-bit low-rank KV cache compression.

## A.1.5. SENSITIVITY TO COMPRESSION WEIGHT

We further analyze the sensitivity of STAR-KV to the compression weight $\gamma$ and the outlier–inlier split ratio. These ablations verify that the reported results are not caused by fragile hyperparameter settings.

| $\gamma$ | Comp (%) | OBQA | PIQA | ARC-e | ARC-c | Hella | Wino | Avg. |
|---|---|---|---|---|---|---|---|---|
| 1.0 | 60 | 42.60 | 76.22 | 71.84 | 42.41 | 68.44 | 66.14 | 61.28 |
| 0.1 | 60 | 41.80 | 75.90 | 72.69 | 41.47 | 70.49 | 66.85 | 61.53 |
| 0.01 | 60 | 42.40 | 76.28 | 71.68 | 42.75 | 68.58 | 66.30 | 61.33 |

*Table 10.* Sensitivity to compression weight $\gamma$ on LongChat-v1.5-32K at 60% KV cache compression.

As shown in Table 10, the average zero-shot accuracy varies only mildly across the tested range. We use $\gamma = 0.1$ in the main experiments because it achieves the best average performance among the tested values.

A.1.6. SENSITIVITY TO THE OUTLIER–INLIER SPLIT

Table 11 shows that the 20:80 split provides a strong balance between compression, speed-up, and downstream accuracy, and is therefore used as the default setting.

| Bits | Out:In | Comp ($\times$) | Speedup ($\times$) | OBQA | PIQA | ARC-e | ARC-c | Hella | Avg. |
|---|---|---|---|---|---|---|---|---|---|
| 16 | – | 1.00 | 1.00 | 41.00 | 76.28 | 71.84 | 41.38 | 71.20 | 60.34 |
| 3.1 | 10:90 | 20.64 | 4.16 | 42.20 | 74.54 | 70.54 | 38.99 | 67.74 | 58.80 |
| 3.2 | 20:80 | 20.00 | 4.09 | 41.80 | 74.86 | 70.83 | 40.70 | 67.72 | 59.18 |
| 3.3 | 30:70 | 19.39 | 4.05 | 42.40 | 74.59 | 71.38 | 38.91 | 67.93 | 59.04 |
| 3.4 | 40:60 | 18.82 | 3.99 | 42.20 | 74.81 | 71.30 | 40.96 | 68.35 | 59.52 |

*Table 11.* Sensitivity to the outlier–inlier split ratio on LongChat-v1.5-32K. The reported average is computed over the listed zero-shot tasks. The speedup is measured on an NVIDIA RTX 4090 GPU.

## A.2. Proof for Lemma 4.2

The feasible set of the joint rank-$R$ decomposition problem is

$$\mathcal{S}_R = \{ Y \in \mathbb{R}^{m \times d} : \text{rank}(Y) \leq R \}, \tag{20}$$

i.e., $\mathcal{S}_R$ is the collection of all matrices with rank at most $R$.

For any head-wise candidate $X = [X^{(1)} \cdots X^{(h)}]$ with $\text{rank}(X^{(i)}) \leq r$ for all $i$, at the same compression budget ($R = hr$), its rank satisfies

$$\text{rank}(X) \leq \sum_{i=1}^{h} \text{rank}(X^{(i)}) \leq hr = R, \tag{21}$$

which means that $X \in \mathcal{S}_R$. Therefore, the entire head-wise candidate family is a subset of the joint candidate family $\mathcal{S}_R$. In particular, $\widehat{W}_{\text{hw}} \in \mathcal{S}_R$ is feasible for the joint optimization, and since $\widehat{W}_{\text{joint}}$ minimizes $\|W - Y\|_F$ over all $Y \in \mathcal{S}_R$, we conclude

$$\|W - \widehat{W}_{\text{joint}}\|_F \leq \|W - \widehat{W}_{\text{hw}}\|_F. \tag{22}$$

## A.3. Training Settings

All experiments are trained using the loss formulation defined in (5), which combines the distillation loss with a compression loss. We use the AdamW optimizer for all experiments. The learning rate is set to $2 \times 10^{-5}$ for all model parameters, while a higher learning rate of $1 \times 10^{-2}$ is used for the adaptive spectral threshold parameters $\alpha$ to enable rapid convergence of rank selection. Training is conducted on a subset of 3,000 samples randomly sampled from the FineWeb-Edu dataset. We first train the model for one full epoch on this dataset using the combined loss. After the adaptive compression parameters converge, we disable the compression loss and continue training using only the distillation loss for an additional 1,000 optimization steps on the same dataset. This second phase further stabilizes the model outputs introduced by compression-aware training. The sequence length is set to 8192 tokens for LongChat-v1.5-7B-32K, LLaMA-3-8B-Instruct and LLaMA-3.1-8B-Instruct models, and 4096 tokens for LLaMA-2-7B. In addition, we do not apply compression to layers 0, 1, and 31, which have been shown to be particularly sensitive across models (Mu et al., 2025), in order to ensure stable accuracy. All experiments are conducted on two NVIDIA RTX PRO 6000 GPUs. Training one epoch under the above configuration takes approximately 6 GPU hours.

## A.4. Benchmark Details

We evaluate the proposed method using three standard benchmark suites: LM-Eval-Harness, LongBench, and RULER.

**LM-Eval-Harness.** We use the LM-Eval-Harness framework to evaluate zero-shot accuracy on a diverse set of reasoning and understanding benchmarks, including OpenBookQA, PIQA, ARC-easy, ARC-challenge, HellaSwag, and WinoGrande. Reported results follow the default evaluation protocols of LM-Eval-Harness, and accuracy (%) is used as the primary metric. For each configuration, we also report the average accuracy across all evaluated tasks.

**Long Context Evaluation.** To assess long-context reasoning and information retrieval capabilities, we evaluate on LongBench tasks, including question answering, summarization, and multi-document understanding benchmarks such as

Qasper, QMSum, TriviaQA, MultiQA, TREC, MultiNews, and VCSum. We report task-specific accuracy or F1 scores as defined by the benchmark, along with the average performance across tasks. We further evaluate long-context robustness using the RULER benchmark. Unless otherwise specified, RULER evaluations are conducted at a fixed sequence length of 4K tokens. We additionally include a 16K RULER evaluation on LongChat-v1.5-32K to test longer-context robustness.

## A.5. Additional Results

### A.5.1. ADDITIONAL MODELS

We include additional evaluations on Mistral-7B-Instruct-v0.2 (Jiang et al., 2023) and LLaMA-2-13B (Touvron et al., 2023) in Table 12. On Mistral-7B-Instruct-v0.2, STAR-KV achieves a higher zero-shot average than ReCalKV at the same 60% KV cache compression rate. On LLaMA-2-13B, STAR-KV substantially outperforms Palu at 60% compression and remains competitive even at 75% compression.

| Model | Method | Comp (%) | Wiki2 | C4 | LM Avg. | OBQA | PIQA | ARC-e | ARC-c | Hella | Wino | Avg. |
|---|---|---|---|---|---|---|---|---|---|---|---|---|
| **Mistral-7B-Inst.** | Baseline | 0 | 5.94 | 9.72 | 7.83 | 46.80 | 80.41 | 81.31 | 55.63 | 83.48 | 74.35 | 70.33 |
| | Palu | 60 | 7.07 | 12.93 | 10.00 | 42.80 | 77.26 | 74.07 | 50.00 | 75.30 | 70.72 | 65.03 |
| | ReCalKV | 60 | – | – | – | 44.00 | 79.27 | 77.78 | 52.20 | 77.88 | 72.30 | 67.24 |
| | STAR-KV | 60 | 8.01 | 10.81 | 9.41 | 44.00 | 80.52 | 79.80 | 52.13 | 78.79 | 71.11 | 67.73 |
| **LLaMA-2-13B** | Baseline | 0 | 5.71 | 8.19 | 6.95 | 44.20 | 79.22 | 77.57 | 50.51 | 79.71 | 71.03 | 67.04 |
| | Palu | 60 | 6.48 | 9.69 | 8.09 | 42.00 | 76.28 | 71.80 | 41.72 | 72.99 | 69.93 | 62.45 |
| | STAR-KV | 60 | 5.69 | 8.22 | 6.96 | 43.60 | 79.00 | 77.36 | 48.12 | 79.30 | 71.27 | 66.44 |
| | STAR-KV | 75 | 5.89 | 8.66 | 7.28 | 42.40 | 77.97 | 75.80 | 46.16 | 77.79 | 68.35 | 64.75 |

*Table 12.* Additional results on Mistral-7B-Instruct-v0.2 and LLaMA-2-13B. For LLaMA-2-13B, perplexity is evaluated at sequence length 1024 due to memory constraints.

### A.5.2. RULER EVALUATION

| Method | Comp (%) | RULER@16K Avg. |
|---|---|---|
| Baseline | 0 | 85.69 |
| Palu | 60 | 62.94 |
| STAR-KV | 60 | 84.16 |

*Table 13.* RULER evaluation at a sequence length of 16K on LongChat-v1.5-32K.

The RULER results in Table 3 show that STAR-KV preserves long-context performance under compression at 4K sequence length. The additional seed on LongChat-v1.5-32K obtains a similar average, indicating stable long-context behavior across repeated calibration runs. At 16K context length, Table 13 shows that STAR-KV remains close to the baseline while substantially outperforming Palu at the same 60% compression rate.

### A.5.3. LONGBENCH EVALUATION

| Method | Comp (%) | Qasper | QMSum | TriviaQA | MultiQA | TREC | MultiNews | VCSum | Avg. |
|---|---|---|---|---|---|---|---|---|---|
| Baseline | 0 | 25.19 | 23.20 | 92.00 | 39.90 | 72.50 | 26.90 | 15.91 | 42.23 |
| Palu | 30 | 14.47 | 23.30 | 86.71 | 26.57 | 73.00 | 26.19 | 8.33 | 36.93 |
| Palu | 50 | 15.38 | 22.10 | 73.36 | 27.58 | 63.50 | 21.66 | 1.95 | 32.21 |
| STAR-KV | 50 | 23.15 | 22.57 | 89.32 | 38.29 | 71.50 | 26.29 | 14.43 | 40.79 |
| STAR-KV | 60 | 22.58 | 22.40 | 81.30 | 41.04 | 65.00 | 25.70 | 15.60 | 39.09 |

*Table 14.* LongBench results on LLaMA-3.1-8B-Instruct.

Table 14 reports additional LongBench results on LLaMA-3.1-8B-Instruct. At 50% compression, STAR-KV achieves an average score of 40.79%, close to the baseline of 42.23% and higher than Palu at 30% compression. Even at 60% compression, STAR-KV remains substantially better than Palu at 50% compression.

### A.5.4. COMBINATION WITH TOKEN EVICTION

STAR-KV compresses KV cache through learned low-rank structure, whereas token pruning methods reduce the number

of retained tokens. These two directions are therefore complementary. Table 15 combines STAR-KV with H2O. On LongChat-v1.5-32K, the combined method with 60% compression from low-rank and 60% pruning, reaches 84% KV cache compression with an accuracy better than H2O (60% pruning) alone. On LLaMA-2-13B, combining STAR-KV with H2O increases compression from 75% (from low-rank) to 90% (combined with 60% pruning) while causing only a small additional zero-shot degradation compared with STAR-KV alone.

| Model | Method | Comp (%) | OBQA | PIQA | ARC-e | ARC-c | Hella | Wino | Avg. |
|-------|--------|----------|------|------|-------|-------|-------|------|------|
| **LongChat-v1.5-32K** | Baseline | 0 | 41.00 | 76.28 | 71.84 | 41.38 | 71.20 | 67.48 | 61.53 |
| | H2O | 60 | 39.20 | 75.63 | 69.99 | 40.36 | 68.99 | 65.51 | 59.95 |
| | STAR-KV | 60 | 41.80 | 75.90 | 72.69 | 41.47 | 70.49 | 66.85 | 61.53 |
| | STAR-KV + H2O | 84 | 40.80 | 75.46 | 70.03 | 40.36 | 68.32 | 64.96 | 59.99 |
| **LLaMA-2-13B** | Baseline | 0 | 44.20 | 79.22 | 77.57 | 50.51 | 79.71 | 71.03 | 67.04 |
| | H2O | 60 | 42.80 | 79.92 | 78.07 | 49.15 | 79.17 | 71.35 | 66.74 |
| | STAR-KV | 75 | 42.40 | 77.97 | 75.80 | 46.16 | 77.79 | 68.35 | 64.75 |
| | STAR-KV + H2O | 90 | 42.20 | 77.97 | 75.46 | 46.33 | 77.67 | 66.85 | 64.41 |

*Table 15.* Combination with H2O. Token eviction and low-rank KV compression are complementary because they reduce different axes of the KV cache.

### A.5.5. LATENCY BREAKDOWN

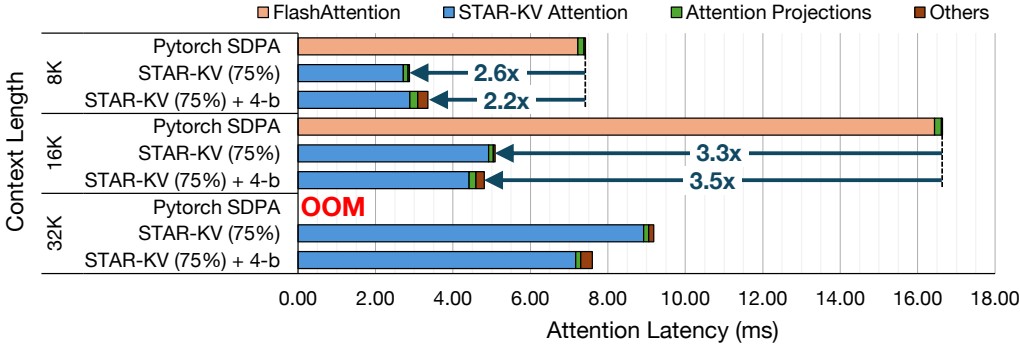

*Figure 12.* The latency breakdown of the attention operator in LLaMA-2-7B on the RTX 4090 GPU. The batch size is 16 in all setups.

We performed a runtime breakdown of STAR-KV at 75% compression with and without quantization on LLaMA-2-7B, comparing it with PyTorch SDPA at context lengths of 8K, 16K, and 32K with batch size 16, as shown in Fig. 12. PyTorch SDPA runs out of memory at 32K, whereas both STAR-KV variants remain executable. At 8K and 16K, STAR-KV reduces KV-cache-related memory-access time by $3.2\times$ and $5.9\times$, respectively, and reduces attention-related computation by $1.9\times$ and $2.4\times$, resulting in overall speedups of $2.6\times$ and $3.3\times$. Adding 4-bit quantization further reduces memory-access time by $4.0\times$ and $8.6\times$, but achieves smaller compute-side gains of $1.1\times$ and $1.6\times$ due to quantization overhead, yielding overall speedups of $2.2\times$ and $3.5\times$. At 32K, STAR-KV without quantization achieves a $4.1\times$ speedup over the estimated baseline, while the variant with quantization reaches $4.9\times$, highlighting the increasing benefit of quantization at longer contexts. These results reveal a clear bottleneck shift: as context length grows, reducing KV-cache memory traffic becomes the dominant source of acceleration, while the relative impact of quantization overhead diminishes.

### A.5.6. SPEEDUP OF ATTENTION ON RTX PRO 6000 GPU

To show the portability of our kernels, we extend the experiments to an RTX Pro 6000 GPU with 96 GB memory capacity. Fig. 13 illustrates the speedup of the attention operator in LLaMA-2-7B, using a batch size 16. Here, we limit the context length and batch size to a value that fits inside the GPU memory with the PyTorch SDPA baseline. At a context length of 128K, STAR-KV achieves a $2.9\times$ speedup, and adding 4-bit KV quantization increases the speedup to $4.3\times$.

### A.6. Implementation Optimizations

**Operation Fusion.** RoPE is often implemented as bandwidth-bound element-wise operations on GPU; therefore, it is common practice to store keys after RoPE in the KV cache to avoid reapplying it repeatedly on the fly. In our setting, the

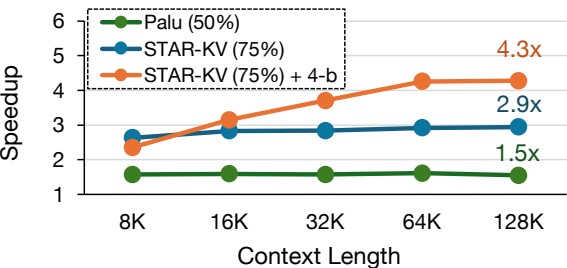

*Figure 13.* Normalized speedup of the attention module over the PyTorch SDPA FP16 implementation in RTX 6000 Pro GPU. The batch size is 16 in all setups.

latent keys are cached in a pre-RoPE form because RoPE can only be applied after reconstructing the latent keys using the decomposed key projections. Therefore, naively applying RoPE to all cached keys at each step would be expensive. Moreover, when the KV cache is quantized, cached keys and values require dequantization before the necessary matrix multiplications. A naive dequantization approach expands low-bit data to higher-precision formats early, increasing memory traffic and incurring additional bandwidth overhead. To address these challenges, we implement custom Triton kernels that fuse (i) key dequantization along with RoPE application, key reconstruction, and attention-score computations $q \cdot K$, and (ii) value dequantization with the reordered attention output computation ($p \cdot V'$). This fusion reduces intermediate materialization and redundant memory reads/writes, improving inference efficiency.

**Attention with GEMM.** Another system-level inefficiency stems from the prevalence of General Matrix–Vector multiplications (GEMV) during decoding, which typically underutilize GPU compute resource compared to General Matrix–Matrix multiplication (GEMM) due to lower arithmetic intensity (Operations/Byte) (Dao, 2023). During decoding, attention forms a single query for the current token against keys/values for the entire context length, leading to multiple GEMV operations across heads. In STAR-KV, the score path ($q \cdot K$) is computed within our fused Triton kernel together with key reconstruction, RoPE, and dequantization, avoiding separate GEMV launches. For the value path, using the attention output reordering in Section 3.3, we stack per-head attention probabilities $p^i \in \mathbb{R}^l$ into $p \in \mathbb{R}^{h \times l}$ and apply them jointly to the shared compressed values $V'$. This transforms $h$ independent GEMV-style operations into a single GEMM, improving throughput by increasing arithmetic intensity and GPU utilization.

