# OpenReview forum: "STAR-KV: Low-Rank KV Cache Compression via Soft Thresholding for Adaptive Rank Control"
_ICML.cc/2026/Conference — ICML 2026 spotlight_

### Official Review · Reviewer_47gK · 2026-02-20

**Soundness:** 3
**Presentation:** 3
**Significance:** 3
**Originality:** 3
**Overall Recommendation:** 4
**Confidence:** 3

**Summary:**

STAR-KV compresses the KV cache in large language models through adaptive low-rank decomposition. The central idea is a differentiable soft-thresholding mechanism applied to the singular values of key/value projection matrices, replacing static or heuristic rank selection. Thresholds are learned per block and per head via a joint objective combining knowledge distillation loss and an adaptive compression loss. Three complementary designs are introduced: (1) a hybrid decomposition applying head-wise decomposition (HD) for keys and joint decomposition (JD) for values, motivated by sensitivity and reconstruction cost analysis; (2) low-rank-aware mixed-precision quantization that partitions latent channels into outlier and inlier blocks at different bit widths; and (3) custom Triton kernels fusing RoPE, dequantization, and reconstruction. Evaluated on four models (LongChat-7B, LLaMA-2-7B, LLaMA-3-8B-Instruct, LLaMA-3.1-8B-Instruct) across perplexity, zero-shot accuracy, LongBench, and RULER, STAR-KV achieves up to 75% low-rank compression (20x with quantization) with minimal accuracy loss and up to 6.9x attention speedup.

**Compliance With Llm Reviewing Policy:**

Affirmed.

**Final Justification:**

Final Recommendation: Weak Accept

The paper presents a well-designed and practically relevant method for KV cache compression using adaptive low-rank decomposition with learned soft-thresholding. The hybrid decomposition strategy and the integration with quantization and efficient Triton kernels are technically sound and well-motivated.

The empirical results are strong, demonstrating consistent improvements in perplexity and significant speedups across multiple models. The method is also practical, with low training cost and clear system-level benefits.

The main limitations include evaluation only on 7B–8B models, limited comparison with complementary approaches such as token eviction methods, and a lack of hyperparameter sensitivity analysis. These issues affect generality but do not undermine the core contribution.

The rebuttal addressed some concerns but did not fully resolve questions regarding scalability to larger models and long-context performance.

Overall, I lean toward weak accept: this is a solid and useful systems contribution that is likely to be adopted, despite some limitations.

Confidence: 4

**Key Questions For Authors:**

1. Do you have any results, even preliminary, on a 13B or larger model? Even perplexity numbers at one compression rate would help assess scalability. If the method scales well, this would strengthen the paper; if the learned rank profiles change qualitatively at larger scale, that would also be informative.

2. Why was RULER evaluated only at 4K tokens? Can you provide results at 16K or 32K? If accuracy degrades sharply at longer sequences under compression, this is important for practitioners to know. If it holds up, that would be a strong selling point.

3. How does STAR-KV interact with token eviction methods like H2O? Can the two be combined (low-rank compression along the feature dimension + token eviction along the sequence dimension)? Even a discussion of whether this is feasible would be helpful.

4. Can you provide a sensitivity analysis for the compression weight gamma and the outlier ratio? The current 20/80 split is fixed across all models and layers; is there evidence this is optimal?

**Limitations:**

The authors acknowledge that GQA-based models pose additional challenges and some layers must be excluded. Not discussed: the limitation to 7B-8B scale, the fixed hyperparameters without sensitivity analysis, the short-context-only RULER evaluation, and potential failure modes on calibration data that is not representative of deployment. These are practical concerns that users of the method would want to understand.

**Strengths And Weaknesses:**

Strengths:

(S1) The soft-thresholding mechanism (Eq. 2) is a clean solution to rank selection. Figure 2 makes the case well: static rank allocation and Fisher-information-based approaches degrade sharply at high compression rates, while the learned thresholds adapt per-head. The formulation using sigmoid with a sharpness parameter s that increases during training (approaching hard thresholding at inference) is a sensible relaxation strategy.

(S2) The hybrid HD/JD decomposition is well-justified through both analysis and experiment. Observation 1 (keys require per-head reconstruction due to RoPE, making HD's lower overhead worthwhile) and Lemma 4.2 with Observation 3 (values are more sensitive to approximation error, favoring JD's lower error) provide a clear rationale. Figure 7 confirms the crossover empirically: HD beats JD for keys, JD beats HD for values, and the hybrid achieves the best of both.

(S3) The results on LLaMA-3-8B-Instruct are strong. At 40% compression, STAR-KV achieves 7.31 perplexity vs. Palu's 7.90 and ReCalKV's 8.08 (Table 2). The combination with 4-bit quantization pushing to 20x compression while maintaining reasonable accuracy (Table 3) is a practical contribution.

(S4) The Triton kernel implementation is a genuine systems contribution. Fusing RoPE with dequantization and reconstruction (Section 5) avoids memory-bound kernel launches. The 6.9x attention speedup and 3.1x end-to-end throughput improvement (Table 4) are measured, not projected.

(S5) Training cost of approximately 6 GPU hours on 3,000 calibration samples (Section 4.3) is practical and makes adoption feasible for most research groups.

Weaknesses:

(W1) All experiments are on 7B-8B parameter models. This is the paper's main limitation. KV cache compression becomes more critical at larger scales (where memory pressure is worse), but the method's effectiveness at 13B, 30B, or 70B is unknown. The learned rank profiles in Figures 10-11 show clear per-layer patterns; do these patterns hold at larger scale, or does the sensitivity landscape change qualitatively? This question is unanswered.

(W2) The baseline set is narrow for a systems paper. Palu and ReCalKV represent the low-rank family, and KIVI/KVQuant the quantization family. But the most popular KV cache compression methods in practice are token eviction approaches like H2O (Zhang et al., 2024) and StreamingLLM (Xiao et al., 2024). These are complementary to low-rank methods (one compresses the token dimension, the other the feature dimension), and a combined approach could be even more effective. The omission of this comparison and discussion is a missed opportunity.

(W3) No hyperparameter sensitivity analysis is provided. The sharpness schedule for s, the compression weight gamma, the 20%/80% outlier/inlier split ratio, and the decision to skip layers 0, 1, and 31 are all fixed without discussion of alternatives. For the layer-skipping in particular, citing Mu et al. (2025) is not sufficient justification; different models may have different sensitivity patterns. A sweep over gamma and the outlier ratio on one model would be informative.

(W4) The RULER evaluation at 4K tokens (Table 5) does not test long-context retention. RULER is designed to evaluate long-context capabilities at sequence lengths of 8K, 16K, 32K, and beyond. Evaluating only at 4K misses the regime where KV cache compression matters most and where accuracy degradation is typically most pronounced. The LongBench results (Table 6) do test longer contexts but show noticeable degradation at 60% compression for LLaMA-3.1-8B-Instruct (39.58 vs. 42.23 baseline), which is not discussed.

(W5) The theoretical contributions (Lemma 4.1, Lemma 4.2) are restatements of standard results (Eckart-Young-Mirsky theorem and subset optimality of singular value truncation). The soft thresholding mechanism, which is the actual novelty, lacks formal analysis. For instance, is the compression loss landscape convex in the threshold parameters? Does the training procedure converge to a unique solution, or are the learned ranks sensitive to initialization?

Minor comments:
- Table 2: Reporting standard deviation across calibration sets would help assess robustness. The perplexity differences between methods at high compression (e.g., 7.31 vs. 7.90) are meaningful, but at low compression the differences are small.
- Section 3.3: The block-wise Hadamard transform applied independently to outlier and inlier channels (Eq. 8) is a reasonable extension, but the justification that "outliers in SVD space concentrate in the top singular-value channels" deserves empirical verification (e.g., a histogram of value magnitudes across channels).
- Figure 8: The learned rank profiles are interesting. An analysis of which layers tend to receive high vs. low ranks, and whether this correlates with known layer importance metrics, would add insight.
- The paper should cite and discuss MiniCache (Liu et al., 2024) and CacheGen (Liu et al., 2024), which also address KV cache compression from different angles.

---

> ### Author Rebuttal · Authors · 2026-03-31
>
> We thank Reviewer 47gK for the detailed analysis and for highlighting the strengths of our work. While
> we try to address all the key questions, some minor points may remain unanswered due to rebuttal space limits, and we are happy to continue the discussion if needed.
>
> ***(Q1/W1) Performance of 13B model***
>
> We include additional results on LLaMA-2-13B, showing that the method maintains strong performance under compression and continues to outperform Palu. We also compare learned rank profiles between 7B and 13B models and observe consistent layer-wise trends, indicating that the sensitivity structure remains largely unchanged at this scale. To account for the differing number of layers, we use mean compression rates across quarters of layers and observe a low mean absolute difference across all quarters **(K: 1.7%, V: 1%)**.
>
> | Model | Comp (%) | wiki2* | c4* | lm-avg* | obqa | piqa | arc-e | arc-c | hella | wino | avg. |
> |-|-|-|-|-|-|-|-|-|-|-|-|
> |LLaMA-2-13B|0|5.71|8.19|6.95|44.20|79.22|77.57|50.51|79.71|71.03|67.04|
> |Palu|60|6.48|9.69|8.09|42.00|76.28|71.80|41.72|72.99|69.93|62.45|
> |STAR-KV|60|5.69|8.22|6.96|43.60|79.00|77.36|48.12|79.30|71.27|66.44|
> |STAR-KV|75|5.89|8.66|7.28|42.40|77.97|75.80|46.16|77.79|68.35|64.75|
>
> *Perplexity is evaluated at 1024 sequence length due to memory constraints.
>
> ***(Q2/W4) RULER at a longer sequence length and accuracy drop on LongBench***
>
> We conduct additional RULER evaluations (mk1, mk2, mq, mv, s1, s2, s3, fwe, sq) at 16K on LongChat-v1.5-32K (60% compression), where STAR-KV outperforms Palu (avg. 84.16 vs. 62.94) with only a small drop from the baseline (85.69 to 84.16). On LongBench with LLaMA-3.1-8B-Instruct, STAR-KV outperforms Palu while providing a higher compression rate.
>
> |Model|Comp (%)|qasper|qmsum|triviaqa|multiqa|trec|multinews|vcsum|avg.|
> |-|-|-|-|-|-|-|-|-|-|
> |LLaMA-3.1-8B-Instruct|0|25.19|23.20|92.00|39.90|72.50|26.90|15.91|42.23|
> |Palu|30|14.47|23.30|86.71|26.57|73.00|26.19|8.33|36.93|
> |Palu|50|15.38|22.10|73.36|27.58|63.50|21.66|1.95|32.21|
> |STAR-KV|50|23.15|22.57|89.32|38.29|71.50|26.29|14.43|40.79|
> |STAR-KV|60|22.58|22.40|81.30|41.04|65.00|25.70|15.60|39.09|
>
> ***(Q3/W2) Combination with H2O***
>
> Thank you for your suggestion. We agree that the two KV cache methods are complementary. The combined approach remains effective, demonstrating the orthogonality of token eviction and low-rank decomposition: on LongChat-v1.5-32K it achieves 84% compression with comparable performance to H2O, and on LLaMA-2-13B it improves compression from 75% to 90% with only 0.34 additional accuracy drop.
>
> |Model|Comp(%)|obqa|piqa|arc-e|arc-c|hella|wino|avg.|
> |-|-|-|-|-|-|-|-|-|
> |**LongChat-v1.5-32k**|0|41.00|76.28|71.84|41.38|71.20|67.48|61.53|
> |H20|60|39.2|75.63|69.99|40.36|68.99|65.51|59.95|
> |STAR-KV|60|41.8|75.9|72.69|41.47|70.49|66.85|61.53|
> |STAR-KV+H20|84|40.8|75.46|70.03|40.36|68.32|64.96|59.99|
> -
> |**LLaMA-2-13b**|0|44.20|79.22|77.57|50.51|79.71|71.03|67.04|
> |H2O|60|42.8|79.92|78.07|49.15|79.17|71.35|66.74|
> |STAR-KV|75|42.40|77.97|75.80|46.16|77.79|68.35|64.75|
> |STAR-KV+H20|90|42.2|77.97|75.46|46.33|77.67|66.85|64.41|
>
> ***(Q4/W3) Analysis on weight gamma and the quantization split ratio***
>
> We perform sensitivity analysis on the compression weight $\gamma$ and the outlier ratio on LongChat-v1.5-32k. The method is robust across a wide range of $\gamma$, with $\gamma=0.1$ giving the best average performance at 60% compression. Varying the outlier:inlier split from 10:90 to 40:60 also shows stable behavior, with 20:80 providing a **strong trade-off between compression, speedup, and accuracy**, which we adopt as the default.
>
> |Gamma|Comp(%)|obqa|piqa|arc-e|arc-c|hella|wino| avg.|
> |-|-|-|-|-|-|-|-|-|
> |1|60|42.6|76.22|71.84|42.41|68.44|66.14|61.28|
> |0.1|60|41.8|75.9|72.69|41.47|70.49|66.85|61.53|
> |0.01|60|42.4|76.28|71.68|42.75|68.58|66.3|61.33|
>
> |Num bits|Out:In Split|Comp(X)|Speedup(X)|obqa|piqa|arc-e|arc-c|hella|avg.|
> |-|-|-|-|-|-|-|-|-|-|
> |16|-|1|1|41.00|76.28|71.84|41.38|71.20|60.34|
> |3.1|10:90|20.64|4.16|42.2|74.54|70.54|38.99|67.74|58.80|
> |3.2|20:80|20|4.09|41.8|74.86|70.83|40.7|67.72|59.18|
> |3.3|30:70|19.39|4.05|42.4|74.59|71.38|38.91|67.93|59.04|
> |3.4|40:60|18.82|3.99|42.2|74.81|71.3|40.96|68.35|59.52|
>
> ***(W5) Convexity of the compression loss and training convergence***
>
> The compression term is $L_{comp}(\alpha)=\sum_l e^{-\alpha_l}$, where each $\alpha_l$ denotes the threshold parameter for layer $l$. Each term satisfies $\frac{\partial^2}{\partial \alpha_l^2} e^{-\alpha_l}=e^{-\alpha_l}>0$, so the Hessian with respect to $\alpha$ is diagonal with strictly positive entries. Therefore, $L_{comp}$ is strictly convex in the threshold parameters. However, the overall training objective (which includes the distillation loss) is non-convex, and thus does not guarantee a unique global optimum. Despite this, we empirically observe stable learned rank profiles across runs and seeds, with only minor deviations. Please refer to our response to Q1 of Reviewer acHp.

---

> > ### Author Rebuttal · Reviewer_47gK · 2026-04-05
> >
> > I have reviewed the rebuttal and appreciate the additional explanations provided.

---

> > > ### Author Response · Authors · 2026-04-08
> > >
> > > We thank Reviewer 47gK for acknowledging our rebuttal, appreciating the additional explanations provided, and maintaining a positive view of our work.

---

### Official Review · Reviewer_Euk6 · 2026-03-05

**Soundness:** 3
**Presentation:** 3
**Significance:** 3
**Originality:** 3
**Overall Recommendation:** 5
**Confidence:** 4

**Summary:**

This paper proposes STAR-KV, a post-training KV cache compression framework that learns fine-grained, adaptive low-rank projections via a differentiable soft-threshold mechanism over singular values, combined with a hybrid decomposition scheme (head-wise for keys, joint for values) and a low-rank-aware mixed-precision quantization using block-wise Hadamard transforms. Empirically, STAR-KV achieves up to 75% KV reduction with negligible accuracy loss and up to 20× overall KV compression when combined with 3.2-bit mixed-precision quantization, while Triton kernels provide up to 6.9× attention speedup and 3.1× end-to-end throughput gains on long contexts. The method establishes new Pareto fronts versus prior low-rank (e.g., Palu, ReCalKV) and quantization baselines (KVQuant, KIVI) across several LLMs and long-context benchmarks.

**Compliance With Llm Reviewing Policy:**

Affirmed.

**Key Questions For Authors:**

What is the memory and storage overhead of keeping per-head U/V factors and Hadamard matrices, and how are these fused in practice to avoid extra reads/writes?

Beyond layers 0,1,31 (which you skip), how stable are the learned rank profiles across seeds/datasets, and do they transfer across nearby model variants (e.g., 7B to 8B)?

**Limitations:**

Yes

**Strengths And Weaknesses:**

Strengths.

STAR-KV presents a well-motivated and fairly complete KV-cache compression stack: (i) learned soft-thresholding to adapt ranks at block/head granularity instead of fixed heuristics, (ii) a hybrid decomposition choice (HD for Keys to reduce reconstruction cost, JD for Values to preserve fidelity) grounded in sensitivity/overhead analysis, and (iii) low-rank–aware mixed-precision quantization (block-wise Hadamard + mixed bits) that leverages the skewed latent-channel statistics after SVD. The paper also reports both quality metrics (PPL, zero-shot, LongBench/RULER) and system metrics, including custom Triton kernels with sizable attention and end-to-end throughput gains, making it more “end-to-end” than many purely algorithmic compression papers.

Weaknesses.

The approach is no longer “pure post-training”: it requires a calibration fine-tuning stage, and its effectiveness may depend on calibration data/domain and tuning details, which are not fully stress-tested for robustness.

Some reported speedups rely on custom Triton fusion and specific baselines/context-length regimes (including OOM-based comparisons), so portability across inference stacks and hardware is uncertain without code.

---

> ### Author Rebuttal · Authors · 2026-03-31
>
> We appreciate Reviewer Euk6 for his positive opinion on our work, and for describing it as a fairly complete KV-cache compression stack.
>
> ***(W1) Stress-testing of the fine-tuning stage***
>
> We thank the reviewer for raising this point. We find the calibration stage robust to both dataset choice and randomness. We ran ablations on C4, RedPajama, and FineWeb-Edu with different seeds on LongChat-v1.5-32K and LLaMA-3.1-8B-Instruct. At fixed 60% compression, variation in average accuracy is very small (**≤0.68%** on LM-Eval), with consistent trends across datasets. Varying the calibration seed also has a negligible effect (**≤0.54** on RULER average), indicating stable behavior across runs. Due to limited rebuttal space, we refer the reviewer to our response to **W2 of Reviewer acHp** for details.
>
> ***(W2) Detailed breakdown***
>
> Thank you for raising this concern. We clarify that **speedup estimation was used only for the PyTorch SDPA baseline**. In contrast, **the latency of STAR-KV at 32K, and STAR-KV + 4-bit quantization at 32K and 64K, was measured directly on GPU**, enabled by the reduced memory footprint. Please refer to our response to Reviewer QtQ7 for additional details on the estimation.
> For portability, our implementation is written in Triton, which was introduced as an intermediate language and compiler for tiled neural network computations [1]. The official Triton repository also documents support for **both NVIDIA GPUs and AMD GPUs via ROCm** [2], indicating portability across supported GPU backends.
> As an additional empirical check, we measured STAR-KV for LLaMA-2-7B on an NVIDIA RTX PRO 6000 Blackwell (96 GB). With batch size 16, STAR-KV achieves **4.3x** speedup at 128K context length.
>
> |Context Length|8K|16K|32K|64K|128K|
> |----------------|-----|-----|-----|-----|------|
> |PyTorch SDPA| 4.64 | 9.01 | 17.41 | 36.16 | 70.3 |
> |STAR-KV (75%) + 4-bit| 1.97 | 2.86 | 4.70 | 8.50 | 16.45 |
> |Speedup (×)| 2.4 | 3.2 | 3.7 | 4.3 | 4.3 |
>
>
> ***(Q1) The overhead of keeping U/V factors and Hadamard matrices, and details of how these matrices are fused***
>
> Hadamard matrices are fused offline into the low-rank factors before inference. For keys, per-head Hadamard matrices are multiplied into the per-head U/V factors; for values, joint Hadamard matrices are multiplied into the joint U/V factors. Since Hadamard matrices are square, fusion preserves factor dimensions and introduces no extra runtime reads or writes.
> During training, there is a modest parameter overhead because we optimize each projection using a full-rank factorization. For a value projection of size $m \times n $ ($n \leq m$), we store $U \in \mathbb{R}^{m \times n}$, diagonal of $S \in \mathbb{R}^{n}$, and $V \in \mathbb{R}^{n \times n}$, which adds $n^2 + n$ parameters relative to the original dense weight. For keys, the head-wise decomposition gives an overhead of $n + n^2/H$ per layer.
> This corresponds to $\sim$501M additional parameters for non-GQA models such as LLaMA-2-7B ($m=n=4096$, $H=32$), or **$\sim$7\%** of the total model size. For GQA models such as LLaMA-3.1-8B-Instruct ($m=4096$, $n=1024$, $H=8$), the overhead is much smaller at $\sim$34M parameters (**$\sim$0.4\%** of total parameters).
> At inference time, there is no additional memory or storage overhead from the Hadamard matrices and U/V factors, because Hadamard matrices are already fused into the low-rank $\tilde{U}$/$\tilde{V}$ factors, truncated according to the learned thresholds, yielding a net reduction in model size. Let $r$ denote the retained rank. We absorb $S \in \mathbb{R}^{r}$ into $U$ and $V$, giving a two-factor form $\tilde{U} \in \mathbb{R}^{m \times r}$ and $\tilde{V} \in \mathbb{R}^{r \times n}$, with a total of $r(m+n)$ parameters. This is smaller than the original $mn$ parameters when **$r < \frac{mn}{m+n}$**. For example, at 60\% compression, we achieve  **20\%** and **50\%**  **lower parameters** for value projection and **60\%** and **60\%** lower parameters for key projection, for non-GQA and GQA models, respectively.
>
> ***(Q2) The stability of the learned rank profiles across seeds/datasets and nearby model variants***
>
> We thank the reviewer for this insightful question. We find that learned rank profiles are highly consistent across datasets and nearby models. On LongChat-v1.5-32K, comparing C4 and RedPajama against FineWeb-Edu, the mean absolute difference is small: for keys, 1.6/1.7 ranks per head out of 128 (**1.2%/1.3%**); for values, 20/7.7 ranks per layer out of 4096 (**0.5%/0.2%**). Across nearby models (LLaMA-2-7B and LLaMA-3.1-8B-Instruct), we compare per-layer compression rates because raw ranks are not directly comparable due to the different dimensions between non-GQA and GQA models. The mean absolute difference is **3.2%** for keys and **11.4%** for values. Overall, the trend is consistent with only minor quantitative variation, indicating that rank profiles are largely preserved across both datasets and closely related model variants.

---

> > ### Author Rebuttal · Reviewer_Euk6 · 2026-04-03
> >
> > Thanks for the rebuttal, my problems has been solved

---

> > > ### Author Response · Authors · 2026-04-08
> > >
> > > We thank Reviewer Euk6 for acknowledging our rebuttal and for their 'Accept' recommendation.

---

### Official Review · Reviewer_QtQ7 · 2026-03-10

**Soundness:** 3
**Presentation:** 3
**Significance:** 3
**Originality:** 3
**Overall Recommendation:** 4
**Confidence:** 3

**Summary:**

This paper addresses the issue of excessive GPU memory and bandwidth consumption caused by key-value caches during long-context reasoning. It proposes an adaptive low-rank compression method based on soft-thresholding, which enables fine-grained control of rank allocation through learnable thresholds. In addition, considering the differences between key and value projections in terms of approximation error and reconstruction cost, the authors design differentiated decomposition strategies. Combined with a low-rank-aware mixed-precision quantization approach and Triton-based custom operator optimization, the method achieves strong accuracy retention and inference acceleration under relatively high compression ratios. Overall, the framework is well-structured, the experimental coverage is broad, and the method achieves a competitive trade-off among compression ratio, model performance, and runtime speed.

**Compliance With Llm Reviewing Policy:**

Affirmed.

**Final Justification:**

My questions have been fully resolved by the authors' response. Consequently, I confirm my initial rating.

**Key Questions For Authors:**

The author mentioned that STAR-KV achieved a 6.9x speedup when using a 64K context length. The article also mentioned that in this case, the PyTorch implementation exceeded memory, so estimated latency was used. Could you explain in detail how the estimation was done? It may not be appropriate to use the estimated acceleration ratio in the conclusion.

**Limitations:**

yes

**Strengths And Weaknesses:**

Strengths
The topic has significant practical relevance. The GPU memory and bandwidth pressure caused by key–value caches in long-context inference is a critical bottleneck in large model deployment. The paper focuses on this issue with a clear objective and strong practical value.

The method design is structurally clear and well-integrated. The soft-threshold-driven adaptive rank control, differentiated decomposition strategies for keys and values, low-rank-aware mixed-precision quantization, and operator-level optimization form a coherent framework rather than a simple combination of techniques. The components are logically connected and mutually reinforcing.

The experiments are relatively comprehensive. The paper reports perplexity, zero-shot task performance, long-context benchmarks, and quantization combination results across multiple mainstream models. It further provides system-level runtime speed and throughput results. Within the evaluated scope, the method demonstrates robust performance in balancing compression ratio and accuracy retention.

The work includes system-level implementation and acceleration validation. The authors not only demonstrate theoretical compression effectiveness but also implement custom operators and report improvements in attention module speed and end-to-end generation throughput, enhancing the practical significance of the work.

Weaknesses
The generalization validation is still mainly limited to medium-scale models. Although representative to some extent, results on larger-scale models or more architectural variants are not yet demonstrated.

There is still room for improvement in the attribution analysis of system acceleration sources. A more detailed breakdown of the contribution of each optimization component and analysis of bottleneck shifts across different scenarios would further strengthen the interpretability of the method.

The related-work positioning could be strengthened. Since the paper focuses on adaptive low-rank KV-cache compression, it would be helpful to explicitly discuss and compare with closely related recent methods such as 《TALE: Token-Adaptive Low-Rank KVCache Approximation with Reconstruction Elimination》.

---

> ### Author Rebuttal · Authors · 2026-03-31
>
> We thank Reviewer QtQ7 for the positive feedback on the importance of the topic, the clear formulation and integration of the method, the comprehensive experiments, and the system-level implementation and validation.
>
> ***(W1) Larger-scale models or more architectural variants***
>
> Our experiments already cover both MHA and GQA, the main architectural distinction relevant to STAR-KV. We further added **LLaMA-2-13B**, a larger model, and **Mistral-7B-Instruct-v0.2**, an additional architecture beyond the LLaMA family. STAR-KV demonstrates superior performance compared to Palu in both models.
>
> | Model | Comp (%) | wiki2 | c4 | lm-avg | obqa | piqa|arc-e|arc-c|hella|wino|avg.|
> |-|-|-|-|-|-|-|-|-|-|-|-|
> | **LLaMA-2-13B** | 0 | 5.71 | 8.19 | 6.95 | 44.20 | 79.22 | 77.57 | 50.51 | 79.71 | 71.03 | 67.04 |
> | Palu | 60 | 6.48 | 9.69 | 8.09 | 42.00 | 76.28 | 71.80 | 41.72 | 72.99 | 69.93 | 62.45 |
> | STAR-KV | 60 | 5.69 | 8.22 | 6.96 | 43.60 | 79.00 | 77.36 | 48.12 | 79.30 | 71.27 | 66.44 |
> | STAR-KV | 75 | 5.89 | 8.66 | 7.28 | 42.40 | 77.97 | 75.80 | 46.16 | 77.79 | 68.35 | 64.75 |
> -
> | **Mistral-7B-Instruct-v0.2** | 0 | 5.94 | 9.72 | 7.83 | 46.80 | 80.41 | 81.31 | 55.63 | 83.48 | 74.35 | 70.33 |
> | Palu | 60 | 7.07 | 12.93 | 10.0 | 42.80 | 77.26 | 74.07 | 50.0 | 75.30 | 70.72 | 65.03 |
> | STAR-KV | 60 | 8.01 | 10.81 | 9.41 | 44.00 | 80.52 | 79.80 | 52.13 | 78.79 | 71.11 | 67.73 |
>
> ***(W2) Detailed breakdown***
>
> We thank the reviewer for the suggestion. We performed a runtime breakdown for STAR-KV and STAR-KV-Q (4-bit KV quantization) at 75% compression on LLaMA-2-7B against PyTorch SDPA at 8K, 16K, and 32K with batch size 16. PyTorch SDPA runs out of memory at 32K, while both STAR-KV variants remain executable. In PyTorch SDPA, memory access accounts for 68.4% of latency on average, versus 29.8% for FlashAttention. At 8K/16K, STAR-KV reduces memory-access time by 3.2x/5.9x and attention-related computation by 1.9x/2.4x, achieving an overall speedup of **2.6x/3.3x**. STAR-KV-Q further reduces memory access by 4.0x/8.6x, with smaller compute-side gains (1.1x/1.6x) due to quantization overhead, yielding **2.2x/3.5x** speedup. At 32K, the speedup of STAR-KV compared to the estimated baseline reaches 4.1x, while it reaches 4.9x for STAR-KV-Q, showing the **larger benefit from quantization**. These results show a clear bottleneck shift: **at longer contexts, reducing KV-cache memory traffic is the main source of acceleration, while the relative impact of quantization overhead diminishes.**
>
> ***(W3) Comparison with TALE***
>
> TALE assigns different ranks and quantization precisions to different tokens based on their importance. It also applies low-rank compression **only to the value states**, rather than to the full KV cache. In contrast, STAR-KV is a framework for full low-rank KV-cache compression. The comparison is summarized below:
>
> | Method | Compression Criteria | Low-rank Comp | Decomposition | Quantization |
> |-|-|-|-|-|
> | TALE | Based on token importance for rank and quantization precision | Only for V | Grouped-head (similar to Palu) | Similar to KIVI |
> | STAR-KV | Based on thresholding singular values for rank | For both K and V | Proposed hybrid KV decomposition | Proposed low-rank–aware mixed-precision quantization |
>
> As TALE has not released code, we compare against the results reported in their paper. We evaluate STAR-KV on LLaMA-3.1-8B-Instruct over the overlapping LongBench tasks. Here, STAR-KV uses 50% low-rank KV compression and 3.2-bit quantization, corresponding to 10x KV-cache compression. On these overlapping tasks, STAR-KV achieves stronger results while operating at a higher overall compression ratio.
>
> | LLaMA-3.1-8B-Instruct | Comp | qasper | qmsum | triviaqa | trec | multinews | Avg. |
> |-|-|-|-|-|-|-|-|
> | TALE | 9.1× | 17.4 | 24.3 | 91.5 | 72.5 | 25.3 | 46.2 |
> | STAR-KV | 10× | 21.81 | 23.42 | 89.32 | 71 | 25.8 | 46.27 |
>
> ***(Q1) The speedup estimation at 64K context length***
>
> We thank the reviewer for raising this concern. Since the PyTorch baseline runs out of memory at 32K and 64K, we could not measure its latency on our hardware. We therefore **estimated the baseline by linear extrapolation** from the measured 8K and 16K latencies. We note, however, that **the latency of “STAR-KV” at 32K and “STAR-KV + 4-b” at 32K and 64K is measured on the GPU**, enabled by their reduced memory footprints. We agree that the extrapolated baseline should be interpreted cautiously, since latency scaling is not strictly linear and tends to grow faster at longer contexts. The estimate should therefore be viewed as **conservative for the speedup and optimistic for the baseline latency**. Please refer to the table below for the measured latencies for LLaMA-2-7B in ms.
>
> | Context Length | 4K | 8K | 16K | 32K | 64K |
> |-|-|-|-|-|-|
> | PyTorch SDPA | 4.25 | 7.40 | 16.64 | OOM (Estimated 37.41) | OOM (Estimated 84.10) |
> | STAR-KV (75%) + 4-b | 2.64 | 3.35 | 4.81 | 7.60 | 12.21 |
> | Speedup (x) | 1.6 | 2.2 | 3.5 | 4.9 | 6.9 |

---

> > ### Author Rebuttal · Reviewer_QtQ7 · 2026-04-02
> >
> > The author's reply solved my problem, and I decided to keep the score.

---

> > > ### Author Response · Authors · 2026-04-08
> > >
> > > We thank Reviewer QtQ7 for acknowledging our rebuttal and keeping their positive view on our work.

---

### Official Review · Reviewer_acHp · 2026-03-12

**Soundness:** 3
**Presentation:** 3
**Significance:** 2
**Originality:** 3
**Overall Recommendation:** 5
**Confidence:** 4

**Summary:**

This paper introduces STAR-KV, a novel approach to KV cache compression that balances memory reduction with model accuracy by combining three core techniques. First, it introduces a differentiable soft thresholding mechanism for different decoder blocks, optimizing a joint objective of adaptive compression loss and knowledge distillation loss to automatically learn the optimal compression rank. Second, based on the finding that joint decomposition (JD) yields less or equal approximation error than head-wise decomposition (HD) but incurs an $h\times$ higher reconstruction overhead, the authors design a hybrid strategy. This strategy applies JD to the more sensitive value projections to preserve accuracy, while adopting HD for the less sensitive key projections to reduce computational costs. Finally, to further reduce the memory footprint, the framework partitions the latent dimension into an outlier block ($Z_{out}$) and an inlier block ($Z_{in}$), applying independent block-wise Hadamard transforms and mixed-precision quantization. Experimental evaluations demonstrate that this combined approach delivers lower perplexity and better downstream performance at higher compression rates compared to existing baselines.

**Compliance With Llm Reviewing Policy:**

Affirmed.

**Final Justification:**

The sensitivity analysis and additional experiments have addressed my concerns and strengthened the paper. Therefore, I have increased my score.

**Key Questions For Authors:**

1. The paper does not analyze the sensitivity of the learnable threshold parameter ($\alpha$) to the calibration data. An ablation study examining how different calibration datasets affect the final compression and performance outcomes would strengthen the evaluation.
2. Could the authors clarify the anomaly on the OBQA benchmark, where the compressed STAR-KV model actually achieves a higher zero-shot accuracy than the uncompressed baseline model?

**Limitations:**

yes

**Strengths And Weaknesses:**

Strength:

1. The paper introduces a differentiable soft thresholding mechanism to adaptively learn the optimal rank for low-rank KV cache decomposition across different attention heads and decoder blocks.

2. The authors propose a hybrid decomposition strategy that treats key and value projections differently based on their sensitivity, followed by a low-rank-aware mixed-precision quantization applied to the resulting latent representations.

3. By implementing custom Triton-based GPU kernels, the framework achieves a 4.0x speedup in the attention operator for a 32K context length.

4. Experimental results show that this approach yields a superior accuracy-compression trade-off, delivering higher overall compression rates while maintaining stronger downstream performance compared to state-of-the-art baselines.

Weakness:

1. The experimental evaluation omits ReCalKV as a comparative baseline for the LLaMA-3-8B-Instruct model in Table 1 and Table 2.

2. The proposed methodology requires a fine-tuning stage on a calibration dataset.

---

> ### Author Rebuttal · Authors · 2026-03-31
>
> We thank Reviewer acHp for the feedback and pointing out the differentiable soft-thresholding mechanism, hybrid decomposition, low-rank-aware quantization, custom Triton kernel implementation, and a superior accuracy-compression trade-off compared to the state-of-the-art baselines as the strengths of our work.
>
> ***(W1) Comparison with ReCalKV for LLaMA-3-8B-Instruct***
>
> We thank the reviewer for raising this concern. Unfortunately, the authors of ReCalKV have not released their implementation, so we were constrained to using only the results reported in their paper. Since their paper does not report results on LLaMA-3-8B-Instruct and LLaMA-3.1-8B-Instruct, we were unable to include ReCalKV as a baseline for those models. To broaden the comparison between ReCalKV and STAR-KV, we have additionally included results on **Mistral-7B-Instruct-v0.2** and **LongChat-7B-v1.5-32k**. Under these settings, **STAR-KV outperforms ReCalKV** on both zero-shot (67.73 vs. 67.24) and long-context tasks (43.4 vs. 40.0).
>
> Mistral-7B-Instruct-v0.2 | Comp (%) | obqa | piqa | arc-e | arc-c | hella | wino | avg.
> ---|---|---|---|---|---|---|---|---
> Baseline | 0 | 46.80 | 80.41 | 81.31 | 55.63 | 83.48 | 74.35 | 70.33
> Palu | 60 | 42.80 | 77.26 | 74.07 | 50.0 | 75.30 | 70.72 | 65.03
> ReCalKV | 60 | 44.00 | 79.27 | 77.78 | 52.20 | 77.88 | 72.30 | 67.24
> STAR-KV | 60 | 44.00 | 80.52 | 79.80 | 52.13 | 78.79 | 71.11 | 67.73
>
>
> LongChat-7B-v1.5-32k (LongBench) | Comp (%) | qasper | qmsum | triviaqa | trec | multinews | avg.
> ---|---|---|---|---|---|---|---
> Baseline | 0 | 27.7 | 22.7 | 82.3 | 68.5 | 26.1| 45.4
> ReCalKV | 60 | 21.0 | 21.1 | 76.1 | 59.0 | 22.7 | 40.0
> STAR-KV | 60 | 22.6 | 22.4 | 81.3 | 65 | 25.7 | 43.4
>
>
> ***(W2) Fine-tuning requirement***
>
> We acknowledge that the proposed method includes a fine-tuning stage on a calibration dataset. However, since the calibration set is small, the associated cost is modest, **requiring less than 6 GPU hours in total** when using NVIDIA RTX Pro 6000 GPU. This step enables finer-grained rank control and, consequently, substantially higher compression during inference. We consider this a small one-time calibration cost that is justified by the resulting inference time efficiency gains.
>
> ***(Q1) Sensitivity analysis of $\alpha$ to the calibration dataset***
>
> We performed additional ablations to study the sensitivity of the learnable threshold parameters to the choice of calibration data. Specifically, we evaluate LongChat-v1.5-32K and LLaMA-3.1-8B using multiple widely used datasets (C4, RedPajama, and FineWeb-Edu with different seeds) for calibration. We observe that the training procedure is **robust to the choice of calibration set**, with only minor variations (up to 0.68%) in final accuracy at the same compression rate of 60% across different datasets.
>
> **LongChat-v1.5-32k**
> | Dataset | Comp (%) | obqa | piqa | arc-e | arc-c | hella | wino | 0-shot avg. |
> |-|-|-|-|-|-|-|-|-|
> | Baseline | 0 | 41.00 | 76.28 | 71.84 | 41.38 | 71.20 | 67.48 | 61.53 |
> | fineweb-edu | 60 | 41.8 | 75.9 | 72.69 | 41.47 | 70.49 | 66.85 | 61.53 |
> | c4 | 60 | 42.2 | 76.77 | 71.76 | 40.78 | 70.53 | 66.3 | 61.39 |
> | RedPajama | 60 | 43 | 76.66 | 71.59 | 41.98 | 68.09 | 65.75 | 61.18 |
> | fineweb-edu (diff seed) | 60 | 42.6 | 76.06 | 71.93 | 41.81 | 68.54 | 65.98 | 61.15 |
>
> **Llama-3.1-8B-Inst**
> | Dataset | Comp (%) | obqa | piqa | arc-e | arc-c | hella | wino | 0-shot avg. |
> |-|-|-|-|-|-|-|-|-|
> | Baseline | 0 | 42.6 | 80.96 | 81.73 | 54.86 | 79.17 | 73.72 | 68.84 |
> | fineweb-edu | 60 | 41.8 | 79.27 | 81.02 | 51.45 | 75.52 | 69.53 | 66.43 |
> | c4 | 60 | 42.8 | 77.97 | 79.55 | 50.94 | 75.1 | 70.09 | 66.08 |
> | RedPajama | 60 | 42.4 | 78.07 | 79.46 | 50.43 | 74.37 | 69.77 | 65.75 |
>
> LongChat-v1.5-32k (RULER @4K) | Comp (%) | mk1 | mk2 | mq | mv | s1 | s2 | s3 | fwe | sq | avg.
> -|-|-|-|-|-|-|-|-|-|-|-
> Baseline | 0 | 99.8 | 99.6 | 98.4 | 96.55 | 100 | 100 | 100 | 48.2 | 56.72 | 88.80
> STAR-KV (diff seed) | 60 | 99.2 | 99.8 | 98.35 | 97.6 | 100 | 100 | 100 | 49.93 | 53.48 | 88.71
> STAR-KV | 60 | 99.4 | 99.6 | 98.5 | 96.55 | 100 | 100 | 99.8 | 46 | 54.45 | 88.26
>
> ***(Q2) Anomaly on the OBQA benchmark***
>
> We appreciate the reviewer for this interesting observation. The variance on OBQA is relatively high (std ~ 2.2) compared to other benchmarks (typically 0.5-1.0), suggesting that performance on this task is inherently more variable. We note that **similar behavior on OBQA is also observed for prior methods** such as Palu [1] and ReCalKV [2]. Additionally, the compression mechanism introduces a mild regularization effect, which can occasionally reduce noise and lead to slight performance gains on certain tasks, as also reported in another KV cache compression work - H2O [3].
>
> [1] Palu: KV-Cache Compression with Low-rank Projection
> [2] ReCalKV: Low-rank Kv Cache Compression via Head Reordering and Offline Calibration
> [3] H2O: Heavy-Hitter Oracle for Efficient Generative Inference of Large Language Models

---

> > ### Author Rebuttal · Reviewer_acHp · 2026-04-03
> >
> > Thanks for the response, my concerns has been fully resolved, I will increase the score.

---

> > > ### Author Response · Authors · 2026-04-08
> > >
> > > We thank Reviewer acHp for acknowledging our rebuttal and increasing their score.

---

### Decision · Program_Chairs · 2026-04-30

**Decision:**

Accept (spotlight)

**Comment:**

This paper attempts to effectively address an important and practical technical challenge of GPU memory and bandwidth usage caused by ky-value caches for long-context reasoning. The proposed approach is clear and has many facets including soft-thresholding for rank adaption, and low-rank aware mixed-precision quantisation. It also provides a system-level contribution in the form of Triton-based custom GPU kernels and strong experimental results. To address some concerns raised by reviewer 47gK regarding larger model sizes and much longer context lengths (upto 32k), some further results were during the rebuttal phase. However, these were still limited to model size upto Llama-2-13B and sequence length 16K. Additional results would have made the case for the paper much more compelling. Despite this, the contribution of the paper remains strong and is positively acknowledged by the reviewers. It is, therefore, recommended to be accepted to the conference.